# Retrospective forecasts of the upcoming winter season snow accumulation in the Inn headwaters (European Alps)

Kristian Förster[1,2,3], Florian Hanzer[3,4], Elena Stoll[2], Adam A. Scaife[5,6], Craig MacLachlan[5], Johannes Schöber[7], Matthias Huttenlau[2], Stefan Achleitner[8], and Ulrich Strasser[3]

[1]Leibniz Universität Hannover, Institute of Hydrology and Water Resources Management, Hannover, Germany
[2]alpS - Centre for Climate Change Adaptation, Innsbruck, Austria
[3]Institute of Geography, University of Innsbruck, Innsbruck, Austria
[4]Wegener Center for Climate and Global Change, University of Graz, Austria
[5]Met Office Hadley Centre, Exeter, Devon, United Kingdom
[6]College of Engineering, Mathematics and Physical Sciences, University of Exeter, Exeter, United Kingdom
[7]TIWAG, Tiroler Wasserkraft AG, Innsbruck, Austria
[8]Unit of Hydraulic Engineering, Institute of Infrastructure, University of Innsbruck, Innsbruck, Austria

*Correspondence to:* Kristian Förster (foerster@iww.uni-hannover.de)

**Abstract.** This article presents analyses of retrospective seasonal forecasts of snow accumulation. Re-forecasts with 4 months lead time from two coupled atmosphere-ocean general circulation models (NCEP CFSv2 and MetOffice GloSea5) drive the Alpine Water balance and Runoff Estimation model (AWARE) in order to predict mid-winter snow accumulation in the Inn headwaters. As the snowpack is a hydrological storage that evolves during the winter season, it is strongly dependent on precipitation totals of the previous months. Climate model predictions of precipitation totals integrated from November to February (NDJF) compare reasonably well with observations. Even though predictions for precipitation may not be significantly more skilful than for temperature, the predictive skill achieved for precipitation is retained in subsequent water balance simulations when snow water equivalent (SWE) in February is considered. Given the AWARE simulations driven by observed meteorological fields as a benchmark for SWE analyses, the correlation achieved using GloSea5-AWARE SWE predictions is $r = 0.57$. The tendency of SWE anomalies (i.e. the sign of anomalies) is correctly predicted in 11 of 13 years. For CFSv2-AWARE, the corresponding values are $r = 0.28$ and 7 of 13 years. The results suggest that some seasonal predictions may be capable of predicting tendencies of hydrological model storages in parts of Europe.

## 1 Introduction

Climate model (CM)-based seasonal predictions are an emerging new field in hydrology (e.g., Yuan et al., 2015; Svensson et al., 2015; Mackay et al., 2015) complementing current progress in predicting long-term developments in changed hydrological conditions as a consequence of anthropogenic greenhouse gas emission. In contrast to climate change projections, seasonal predictions focus on hydrological states of the upcoming months from their dependence on initial states (Warner, 2011). These can provide "Climate Services": A set of tools, products and information, serving decision makers and practitioners and bringing all types of information on climate research into practice at all levels of society (Vaughan and Dessai, 2014).

This makes them relevant for detecting anticipated short-term changes in hydrological systems as requested by international research programmes such as "World Climate Research Programme" (WCRP, e.g., Kirtman and Pirani, 2009), "Future Earth" (Greenslade and Berkhout, 2014), and more specialised programmes like, e.g., the International Network for Alpine Research Catchment Hydrology (INARCH, Pomeroy et al., 2015) which is part of the Global Energy and Water Cycle Exchanges Project (GEWEX, Chahine, 1992). In this context, seasonal predictions contribute, on the one hand, to cope with the WCRP "Grand Challenges" (Reid et al., 2010), and are used on the other hand to detect short-term changes in coupled hydrological-societal systems. The latter consideration of water and humans (Sivapalan et al., 2012) seeks to better understand interactions between society and hydrological systems for which seasonal predictions can also be seen as relevant. The goal of the current scientific decade "Panta rhei" of the International Association of Hydrological Sciences (IAHS) is to better understand these interactions on different scales of time (Montanari et al., 2013).

Seasonal outlooks of hydrological variables have been prepared for decades. Antecedent hydrological and meteorological data have been used to predict monthly to seasonal streamflow using statistical methods (e.g. regression models) in various hydrological regimes (Pagano et al., 2004; Robertson and Wang, 2012; Schick et al., 2015). Another common way to predict future hydrological states is to run a process-based hydrological model based on known initial states and force it with ensembles of meteorological data observed in the past. This methodology is well known and referred to as Ensemble Streamflow Prediction (ESP, Wood and Lettenmaier, 2008). The development of this method goes back to the seventies and eighties (Twedt et al., 1977; Day, 1985) and framed the development of statistical seasonal hydrological forecasting. ESP is a very useful method to study the influence of meteorological boundary conditions, which are obtained from observed long-term records, on the results of the hydrological forecasting model. In contrast, the reversed-ESP experiment is based on actual meteorological forcing but involves an ensemble of initial states, which makes it an appropriate method to study the influence of initial conditions on forecast results. The combination of both methods is also subject to recent research on predictability of hydrological systems (e.g. the VESPA approach, Wood et al., 2016). In the last decades, coupled Atmosphere-Ocean General Circulation Models (AOGCM) have become a viable method for seasonal predictions (Svensson et al., 2015). These climate model-based forecasts provide future meteorological/climatological conditions for the next weeks (sub-seasonal forecasts), months (seasonal forecasts), or decades (decadal forecasts) on a physical basis rather than on statistics. An overview about the state-of-the-art of CM-based seasonal predictions is provided by Doblas-Reyes et al. (2013) and Yuan et al. (2015). Like numerical weather prediction, seasonal forecasts are on the one hand an initial-state problem since predictions of the atmospheric states of the upcoming months strongly depend on the initial states of the atmosphere, oceans, land, and sea ice. In contrast to weather predictions, the need for considering ocean and sea ice dynamics is crucial since these components of the climate system affect atmospheric phenomena on time scales beyond typical weather predictions. Another important difference from numerical weather predictions is the dependence of seasonal predictions on boundary conditions. Like long-term climate predictions which are based on anthropogenic greenhouse gas emissions, CM-based seasonal predictions require adequate definitions of boundary conditions (Doblas-Reyes et al., 2013).

The skill of CM-based seasonal predictions is not distributed equally in space and time (Smith et al., 2012; Kim et al., 2012; Kirtman et al., 2014). For instance, the skill in Europe is much lower than in the tropics, where phenomena like El

Niño Southern Oscillation (ENSO) are predictable with higher accuracy (Yuan et al., 2015). Current progress on improving predictability has been recently reported by Scaife et al. (2014) who demonstrated skilful predictions of the North Atlantic Oscillation, a feature that is relevant for seasonal predictions in Europe. Bruno Soares and Dessai (2015) found that there is a mismatch in supply and demand regarding seasonal forecast products which is limited by skill levels in some regions although the authors also detected additional non-scientific reasons for this mismatch like, e.g., insufficient communication of forecasts to the users.

In general, hydrological forecast models are quite sensitive to initial hydrological conditions such as antecedent rainfall, soil moisture and SWE. Uncertainties in the data of antecedent meteorological conditions influence the quality of process-based hydro-meteorological models in hourly resolution, e.g. in case of two days flood forecasts (Achleitner et al., 2012) or one month sub-seasonal streamflow drought forecasts (Fundel et al., 2013). Statistical seasonal streamflow forecast models can be improved when initial conditions with respect to soil moisture and groundwater flow (Robertson et al., 2013) or snow water equivalent (Pagano et al., 2004) are considered. Discharge from alpine catchments is known to be related to snow and ice melt (Viviroli et al., 2003; Kaser et al., 2010). For hydro power generation it is interesting to know if a winter season is above or below average regarding the accumulation of snow. For water management demands such as efficient hydro power production, large efforts have been made to measure SWE in catchments of reservoirs (Painter et al., 2016; Krajči et al., 2017; Schattan et al., 2017), to simulate distributed SWE in basins of reservoirs and water in-takes (Schöber et al., 2012; Hanzer et al., 2016), to improve flood forecasts with distributed SWE data (Schöber et al., 2014) and to model future runoff under climate change conditions in snow and ice melt dominated catchments (Barnett et al., 2005; Schaefli et al., 2007; Finger et al., 2012; Hanzer et al., 2017). Gridded SWE data used for initialization of a process-based hydrological model improved predictions of SWE with lead times up to one month (Jörg-Hess et al., 2015). Seasonal streamflow and reservoir inflow predictions in snow-dominated basins were quite skillful during the snow-melt season and showed larger uncertainties during the rest of the year (Schick et al., 2015; Anghileri et al., 2016).

Besides hydro power, seasonal prediction of the accumulation of snow may be relevant to estimate the future evolution of snow depth on skiing slopes for the winter tourism business (Abegg et al., 2013; Marke et al., 2015). Well-focused, sustainable operation of technical snow production could be a means of significant savings with respect of energy costs and water use (Hanzer et al., 2014; Olefs et al., 2010).

In the present study, we focus on a way to make seasonal hydrological predictions better exploitable for demands in context of water resources planning in the Alps. We present a systematic evaluation in order to detect the predictability of above and below average snow accumulation which is expected to significantly influence runoff in spring and early summer. To achieve this goal, CM-based seasonal forecasts are employed as input data to a water balance model that predicts snow water equivalent (SWE) and runoff in the Inn headwaters. A new aspect of this work is the focus on hydrological storages instead of instantaneous hydrological fluxes and the seasonal prediction of SWE in general. It is expected that the focus on integrated storages (e.g., mid-winter snow accumulation) is more robust than considering instantaneous fluxes (e.g., precipitation, runoff) in seasonal predictions.

Moreover, we focus on the winter season as extratropical seasonal forecasts appear to have the highest skill in this season (e.g. Riddle et al., 2013; Scaife et al., 2014; Kang et al., 2014). There are a number of reasons for this, including winter being the season when the stratosphere is active, which is known to affect predictions (e.g. Domeisen et al., 2015; Scaife et al., 2016; Butler et al., 2016). The winter season also shows much stronger dynamical connections to the tropics, allowing high predictability of tropical rainfall (Kumar et al., 2013) to be transmitted into the extratropics (Greatbatch et al., 2012; Molteni et al., 2015; Scaife et al., 2017).

Based on that, the research question remains: Can we detect above or below average snow conditions based on CM-based seasonal predictions in the Alps? To answer this, CM-based and hydrological modelling is applied in an alpine case study. In Section 2 the relevant information about the study area, the climate data, the CM-based seasonal predictions, the water balance model, and the methodology for detecting the predictability of snow accumulation are described. In Section 3, the results are presented, compiled and discussed. Finally, Section 4 provides concluding remarks and an outlook for future work.

## 2 Material and methods

### 2.1 Study area

The Inn headwaters catchment upstream of the Kirchbichl gauging station covers an area of $9,310\,\text{km}^2$ and is located in Switzerland and Austria (see, Fig. 1). The Inn river is the main tributary to the upper Danube. Elevations in the catchment range between 486 and 4049 m a.s.l., with a mean elevation of approximately 2000 m a.s.l. About 3% of the catchment area is covered by glaciers. During the winter season runoff is lowest since a major fraction of precipitation is accumulated to the snow cover. In spring snow melt causes an increase in runoff reaching its maximum in August when glacier melt is highest. For the period 1985–2009, the average areal precipitation and runoff amount to $1225\,\text{mm}\,\text{yr}^{-1}$ and $1000\,\text{mm}\,\text{yr}^{-1}$, respectively. In the second half of the 20th century, several reservoirs have been built in the study area. Their capacity in total is $638 \cdot 10^6$ m$^3$.

### 2.2 Climate data and seasonal predictions

#### 2.2.1 Climate data

The climate data provided by the HISTALP project (Historical Instrumental Climatological Surface Time Series of the Greater Alpine Region, Auer et al., 2007) is a suitable dataset for studies on climatology and long-term changes of temperature and precipitation in the Alps. The data has been compiled for a long period of time (1800-2010) and includes a dense observational station network from different countries in the Greater Alpine Region. Moreover, it has been quality checked and homogenised (Auer et al., 2007; Chimani et al., 2013). Mean temperature and precipitation depth are provided on a grid with a temporal resolution of one month and a spatial resolution of five minutes (approx. 6 km).

### 2.2.2 Climatological forecasts

In the framework of this study, the term "climatological forecasts" refers to simulations based on long-term averages of air temperature and precipitation depth for each month based on the HISTALP data. For instance, considering a climatological forecast for January, mean air temperature and precipitation depth are computed through averaging each variable over all Januaries in a multi-year period (i.e. 1996–2009).

### 2.2.3 Climate model-based seasonal predictions

In this study, two different AOGCMs are utilised as input data for further analyses of seasonal predictions. As outlined earlier, the requirements of CM-based seasonal predictions exceed the extent of numerical weather predictions with respect to the forecast horizon and the number of subsystems of the climate system that need to be considered. Due to the extended forecast horizon, oceans and sea ice need to be incorporated in the models as well (see, e.g., Smith et al., 2012; Doblas-Reyes et al., 2013; Yuan et al., 2015). In this study, two different AOGCMs are applied independently:

– NCEP's (National Centers for Environmental Prediction) Coupled Forecast System model version 2 (CFSv2, Saha et al., 2014) is an operational seasonal prediction system. Forecasts are initialised four times a day. The horizontal resolution is 0.5°(approx. 40 km). In order to derive monthly forecasts, runs between the 8th day of the previous month and the 7th day of the current month are utilised in order to generate a lagged ensemble. This methodology is proposed by Yuan et al. (2013) who applied this method to re-forecasts. Since re-forecasts are only available for every 5th day, a typical ensemble of CFSv2 re-forecasts comprises 24 members per month. The archive of re-forecasts includes data from 1985 to 2009. The maximum lead time is 9 months.

– MetOffice Global Seasonal forecast system version 5 (GloSea5) is a seasonal prediction system that runs operationally at the MetOffice (MacLachlan et al., 2015; Scaife et al., 2014). Compared to CFSv2, it has a higher ocean horizontal resolution (0.25°, approx. 20 km). The data applied in this study was provided by the SPECS project ("Seasonal-to-decadal climate Prediction for the improvement of European Climate Services", http://www.specs-fp7.eu/) and covers the period between 1996 and 2010. Re-forecasts for winter were used with initial start dates: 25 Oct., 01 Nov., and 09 Nov. For each date, 3 runs are available which gives a lagged ensemble of 9 members per winter. This subset of hindcasts has a lead time of 4 months for each run.

Systematic analyses are performed for 1996 – 2009 (the period in which both models are available). Only those re-forecasts that start in November are considered. The lead time is limited to 4 months to predict snow conditions in February. Monthly grids of the climate models with their original grid spacing (as specified above) are used as forcing data for the water balance model which is described in the next section.

## 2.3 Water balance simulations using AWARE

The Alpine Water balance and Runoff Estimation model (AWARE, Förster et al., 2016) is a deterministic hydrological model operating on a regular grid at one month time steps. The model has been designed to estimate anomalies in hydrological variables at the catchment scale from anomalies in meteorological fields predicted by climate models. The coarse temporal resolution allows one to carry out seasonal predictions considering a large number of individual runs at a minimum of computational costs which justifies the coarse time step. As the study's focus is on anomalies in seasonal characteristics, using a monthly scale water balance model is feasible (Kling et al., 2012; Bock et al., 2016) and these models are also applied for seasonal hydrological predictions (Bell et al., 2017).

Required meteorological forcing data include both mean monthly air temperature and monthly precipitation totals provided as grids or station data, which makes the model parsimonious with respect to data requirements. Altitudinal gradients are applied in order to realistically redistribute temperature and precipitation on the model grid. In general, this feature results in a decrease in temperature with increasing elevation and an increase in precipitation on the mountains. For each grid cell the relative contributions of rainfall and snowfall are computed taking into account two threshold temperature values. If the air temperature falls below the lower threshold temperature, the monthly precipitation depth is assumed to be snowfall only. In contrast, air temperatures exceeding the upper threshold indicate rainfall only. In order to enable the occurrence of both snow and rain, a transition range between both thresholds is defined. Based on air temperature, the fraction of rain and snow is linearly interpolated between these two thresholds. Even though the model is also capable of reading shortwave radiation fields (Förster et al., 2016) in order to improve ice melt predictions, a simplified snow- and ice-melt simulation using air temperature only is possible. This simplification considers the fact that air temperature and precipitation are readily available and more predictable compared to some other meteorological fields. In order to perform simulations with this minimal input of data, the Thornthwaite (1948) evapotranspiration approach is applied. The soil water balance is computed following the approach of McCabe and Markstrom (2007). A linear storage is applied in order to account for the recession of runoff typically related to groundwater processes.

The spatial resolution of the Inn headwaters setup in the AWARE model is 1000 m. Besides a grid based model domain, AWARE assumes a baseline (reference) meteorological dataset for calibration, which is shown in Figure 2 using the HISTALP data from 1996 to 2009 as the reference period (this run is herein referred to as HISTALP-AWARE). The Nash-Sutcliffe model efficiency (NSE) amounts to $E = 0.92$ which might be viewed as very good model performance. As suggested by Schaefli and Gupta (2007), the benchmark Nash-Sutcliffe model efficiency is computed as well ($E_b = 0.45$). This benchmark NSE value accounts for strong effects of seasonality (Eq. A1 in the Appendix Sect. A). While the standard NSE indicates if a model is better than the average of observed values, the benchmark NSE proves if the model performance exceeds the corresponding value of a simple model that simply predicts long-term averages for each month. Since the benchmark NSE is also greater than zero, the model is more skillful than applying long-term averages. According to Klemeš (1986) a split sample test is applied including an independent validation period ranging from 1984 to 1995. The corresponding NSE and benchmark NSE are $E = 0.91$ and $E_b = 0.25$, respectively. A possible reason for the lower $E_b$ value might be the fact that the validation period

has seen an advancing of glaciers due to positive glacier mass balances. In contrast, the calibration period is characterised by a shrinkage of glaciers volumes. Both processes are not incorporated in the model so far. However, as the model performance of the validation period is still comparable to the calibration period, the model is found to be suitable for predictions. The mismatch of runoff simulations in winter, especially in March can be attributed to the effects of reservoirs on river flow in the catchment area which are not represented in the model so far. In this period water is released from seasonal storages which are filled in summer.

Another advantage of the one month time step is the lower complexity with respect to downscaling of climate model data. Current approaches focus on statistical (e.g., Crochemore et al., 2016) or dynamical downscaling (e.g., Förster et al., 2014) of coarse atmospheric data fields (e.g., derived by climate models). AWARE builds upon a simple and robust approach which is based on anomalies. For instance, Marzeion et al. (2012) successfully add anomalies from other datasets to a reference climatology to compute glacier mass balances at the global scale. In order to account for different spreads of distributions, standardised anomalies are considered in our study. According to Wilks (2006) this approach is feasible when "working simultaneously with batches of data that are related, but not strictly comparable". This is a typical situation in case of observational data and re-forecasts. Standardised anomalies $z_x$ are simply computed for a variable $x$, taking into consideration its long-term mean $\bar{x}$ for a given month and the corresponding empirical standard deviation $\tilde{s}_x$ (Wilks, 2006):

$$z_x = \frac{x - \bar{x}}{\tilde{s}_x} \tag{1}$$

Given that two datasets $x$ and $y$ are comparable (e.g., reference climatology and the climatology of re-forecasts), their standardised anomalies $z_x$ and $z_y$ might be viewed comparable as well. Based on the assumption that $z_x = z_y$, Eq. 1 can be rearranged to

$$x = \frac{y - \bar{y}}{\tilde{s}_y} \cdot \tilde{s}_x + \bar{x}. \tag{2}$$

Anomalies of the climate model (i.e. $y - \bar{y}$) can easily be transformed to the climatology of the reference data set (i.e., $x$). Mean values and standard deviations are computed separately for each month and climate data set including HISTALP, CFSv2, and GloSea5. In this way, anomalies predicted by the climate models can be reliably transformed to typical anomalies of the observational data.

## 2.4  Model experiment for analysing the predictability of snow accumulation

The long-term simulations of the water balance provide monthly snapshots of valid system states for each state variable at any point in time. For each CM-based seasonal prediction run starting in November, system states for SWE, soil moisture, and groundwater storage computed for October are defined as initial states. In total four AWARE runs driven with different forcing datasets are available for each winter season between 1997 and 2009 (November to February, NDJF):

1. HISTALP-AWARE: Long-term continuous run based on observed HISTALP data (see, Sect. 2.2.1 and Fig. 2),

2. CF-AWARE: Climatological forecasts (CF) with average conditions computed using HISTALP (see, Sect. 2.2.2),

3. GloSea5-AWARE: CM-based seasonal forecast using GloSea5 (ensemble mean of 9 members), and

4. CFSv2-AWARE: CM-based seasonal forecast using CFSv2 (ensemble mean of 24 members).

The ensemble provided by each CM-based seasonal forecast of meteorological quantities is averaged prior to the water balance simulations. In general, ensemble seasonal predictions are subject to low signal to noise ratios. The signal in the ensemble mean is small in most cases and using members individually will mask the signal (Scaife et al., 2014; Eade et al., 2014). In general, each ensemble member of input data is individually processed in hydrological forecasting, which is why the averaging is typically implemented afterwards. However, a skill improvement is reported by recent seasonal prediction studies (e.g., Bell et al., 2017) in which the concept of averaging is applied prior to hydrological simulations. This approach seems feasible given that the time step of hydrological simulations is one month. Although the hydrological model is a conceptual model that mimics the basic physical principles, the temporal scale does not allow to capture the full dynamics of hydrological processes that are typically studied on smaller scales. Thus, the coarse temporal resolution of the modelling approach is to a certain degree "statistical" in nature which justifies the application of mean ensemble inputs. Moreover, the utilisation of standardised anomalies applied to CM-based seasonal forecasts in the AWARE model accounts for variance corrections to the ensemble mean values as suggested by Eade et al. (2014). Appropriate uncertainty can also be added to the predictions to ensure reliable probabilistic forecasts.

The basin-average time series of these water balance simulations are directly comparable. While the continuous long-term simulation represents a reference run (#1) serving as benchmark for seasonal predictions, the climatological forecasts (#2) help to judge whether anomalies will be above or below average. Correlations between the reference run and the water balance simulations forced by CM-based forecasts (#3 and #4) are computed to assess the predictive skill. Moreover, the tendency or sign of anomalies is compared through counting the coincidence of above (below) average anomalies in the reference run and the seasonal predictions.

A set of skill measures is used throughout the study in order to quantify the model skill of the different forecasts (CF-AWARE, GloSea5-AWARE, CFSv2-AWARE). Besides correlation and hitrate (i.e. the number of correctly predicted states divided by the total number of winters) other measures to assess the skill of the models are considered. For instance, the standard deviation of a single time series is a measure to compare the variability of forecasts. In contrast, the Root Mean Square Error (RMSE) also involves observed time series and gains insight into the absolute difference between time series. Since quadratic differences are summarised, a greater weight is assigned to larger differences, thus making RMSE sensitive to greater mismatches. In order to show the accuracy of the models for predicting the tendencies of anomalies (hitrate), the Brier Skill Score (BSS) is also computed (see Eq. A2 and Eq. A4 in Appendix Sect. A along with a brief description). In general, a skill score judges the improvement of a forecast system relative to a reference (climatology). A value of zero would indicate that the forecast system is not better than the reference. In contrast, a value of one indicates a perfect match of forecasts and observations. The BSS is related to the hitrate which has already been defined (higher hitrates go in hand with higher BSS). Finally, the Mean Absolute Error Skill (MAE, Eq. A3) is comparable to RMSE but does not account for quadratic weighting

of differences. Like BSS, it can be computed as skill score MAESS (Eq. A4) that is a measure for the differences in absolute terms. In this way, it is less sensitive than RMSE to large differences but rather includes a reference run.

## 3 Results and Discussion

### 3.1 Long-term simulations and climatological forecast of SWE

While the applicability of AWARE to reconstruct the water balance in terms of observed runoff time series was demonstrated in Sect. 2.3, it is necessary to evaluate the model experiments HISTALP-AWARE and CF-AWARE with respect to SWE prior to the analyses of CM-based SWE forecasts. Figure 3(a) demonstrates the annual cycle of modelled SWE. The black dashed line is the mean value of all years computed using the reference run (HISTALP-AWARE). It compares well with the black bold line which represents the climatological simulations based on AWARE using average air temperature and precipitation
depth for each month (CF-AWARE). Thus, a climatological forecast is suitable to compute average snow conditions. Figure 3(c) shows the spatial distribution of average SWE in February. The averages of SWE on the model highlights the typical snow distribution with highest values on the mountains and lower values in the valleys. Full time series are shown in Fig. 3(b). The boundary conditions of the climatological forecast are equal in each year. However, the initial conditions differ according to the initialisation each year in October which is obtained from the long-term run. Figure 3(d) depicts a subset of SWE observations
compiled by Schöber et al. (2016). In contrast to the cited study, which explains the methodology of SWE sampling in detail, here only stations above 1400 m a.s.l. have been selected in order to better match the average catchment elevation (Sect. 2.1). The correlation between computed SWE in February and the SWE observations in February is $r = 0.65$ (Fig. 3(b) vs. Fig. 3(d)). This comparison should be interpreted with caution. First, despite the fact that a sub-selection of stations that better match the mean elevation of the catchment has been chosen for this analysis, the full range of elevation bands in the basin
is not fully covered by the observational dataset. Moreover, scaling issues limit spatial and temporal representativeness, since averaged point-scale measurements recorded on a weekly scale are compared to basin-scale water balance simulation with one month time step. However, observed and computed SWE compare reasonably well. This underlines the applicability of AWARE to predict SWE.

### 3.2 CM-based seasonal predictions using AWARE

In a next step, anomalies computed using AWARE forced by CM-based seasonal forecasts are compared to the corresponding values of the reference run (HISTALP-AWARE, #1). This evaluation is demonstrated in Fig. 4 for temperature, precipitation depth, and SWE in February. Anomalies in temperature and precipitation depth refer to the period November to February (NDJF) in each winter and represent average values at the basin scale (i.e. the mean of all grid points of the meteorological fields in AWARE). In this way, the values are subject to the statical transformations and elevation dependent redistributions as
outlined in Section 2.3. The anomalies of the reference AWARE run driven by HISTALP are shown in the top panels of Fig. 4 (HISTALP-AWARE). Their correlation is set to 1 by definition since this run is viewed as reference. The seasonal forecasts

computed using AWARE driven by GloSea5-AWARE (center) and CFSv2-AWARE (bottom) are also displayed. In addition, Tab. 1 (first model experiment column) provides a summary of skill measures for temperature, precipitation, and SWE.

Correlation coefficients computed for NDJF temperature anomalies range from $r = 0.17$ (CFSv2-AWARE) to $r = 0.32$ (GloSea5-AWARE). Tendencies in anomalies (i.e. the prediction of correct signs of anomalies) also vary between the models. This becomes obvious when counting the shaded areas indicating a mismatch between the seasonal forecast and the reference run. While GloSea5-AWARE correctly predicted the sign of temperature anomalies in 9 of 13 winters, the hitrate achieved for CFSv2-AWARE only amounts to 8 of 13 (see, Tab. 1). The differences between GloSea5-AWARE and CFSv2-AWARE in terms of standard deviation are small. Hence, both model settings show a similar variability of forecasts which can be attributed to the standardised anomaly approach. GloSea5-AWARE shows a smaller RMSE than CFSv2-AWARE does. A similar ranking of skill is obvious when considering the skill scores BSS and MAESS. The latter suggests that both model runs (GloSea5-AWARE and CFSv2-AWARE) are less skilful than climatology (MAESS < 0). However, the positive BSS values highlight the capability of predicting the tendency of temperature anomalies.

In the case of GloSea5-AWARE, the hitrate of correctly predicted anomalies regarding precipitation is 9 of 13 ($r = 0.61$). As for temperature, the model skill of precipitation predictions computed by CFSv2-AWARE is also lower (hitrate 7 of 13, $r = 0.31$). This finding holds also true for the other skill measures, namely RMSE, BSS, and MAESS. However, the number of correctly predicted tendencies achieved using GloSea5-AWARE might be viewed as good results since the seasonal forecasts includes a lead time of 4 months. Single months show lower scores, suggesting that a temporal integration improves the robustness of results consistent with our approach using hydrological storages rather than fluxes. In our study, we found monthly correlations computed for precipitation forecasts ranging from -0.29 to 0.30 (GloSea5-AWARE) and -0.11 to 0.15 (CFSv2-AWARE), respectively. These are generally lower than the corresponding values achieved for the averaged NDJF forecasts (GloSea5-AWARE: 0.61, CFSv2-AWARE: 0.31). Similar values of the same order have been observed for SWE forecasts (GloSea5-AWARE: 0.57, CFSv2-AWARE: 0.28).

Given the skill measures from Tab. 1 (first column) and the coincidence of anomalies highlighted in Figure 4(c) the predictive skill achieved for precipitation depth is also prevailing for SWE in February. Even though correlation coefficients are slightly lower compared to precipitation depth (GloSea5-AWARE: $r = 0.57$, CFSv2-AWARE: $r = 0.28$), SWE values in February computed by AWARE driven by CM-based forecasts compare well to those of the reference run (HISTALP-AWARE). The hitrate achieved using GloSea-AWARE even reaches 11 of 13 while the hitrate of CFSv2-AWARE remains at the level of 7 of 13. An increase in skill in terms of RMSE, BSS, and MAESS is also obvious - at least partially - for both models indicating that some skill measures suggest that SWE predictions are more robust than precipitation predictions.

A Bernoulli experiment helps to judge whether these hitrates differ from the performance of a "fair coin" for predicting above and below average conditions. The null hypothesis is: The hitrate of the seasonal forecasts does not differ from a random 50:50 probability (binomial test). Given the total number of winters $n = 13$ and a level of significance of $\alpha = 0.05$, the null hypothesis is rejected for hitrates above 9 of 13. This means that according to the results shown in Fig. 4 and Tab. 1, for seasonal predictions of SWE using GloSea5 this test rejects the null hypothesis, indicating significant skill. In contrast, the scores for CFSv2 are not significant.

Regardless the limitations discussed with respect to observed SWE, the correlations are much lower if the observations from Fig. 3(d) are involved in skill computations. The correlation between observed anomalies and GloSea5 is $r = 0.21$ while the corresponding value achieved using CFSv2 is only $r = 0.11$. These values are much lower than the correlations achieved using the reference run (HISTALP-AWARE). This finding might also be related to possible mismatches in representativeness between observations and simulations. However, the comparison between HISTALP-AWARE and the CM-based seasonal forecasts highlights GCM-forecast skill and acknowledges the fact that the water balance model is never perfect since it introduces uncertainties into hydrological forecasts, too. Due to the reasonably good agreement between seasonal forecasts and the reference run, the skill of CM-based forecasts is viewed promising.

Figure 5 depicts time series of the water balance of the snow storage for each year and each AWARE model run. Monthly precipitation (divided into rainfall and snowfall), cumulative snowmelt, and SWE are plotted. Moreover, the snow accumulation of the reference run (HISTALP-AWARE, #1) and the climatological forecast (CF-AWARE, #2) are displayed. The latter is subject to the same forcing in each year but is initialised according to the system states of AWARE in late fall. If the SWE computed by HISTALP-AWARE exceeds the corresponding value of CF-AWARE, above average snow accumulation prevails. Accordingly, the opposite is true for below average conditions. A similar comparison is possible for the predictions of GloSea5-AWARE and CFSv2-AWARE. If the CM-based forecast and HISTALP-AWARE simultaneously indicate either above or below average conditions, a label "HIT" is added to the corresponding seasonal forecast. The overall hitrate is readable from Tab. 1. Even though monthly precipitation depth differs between HISTALP-AWARE and the CM-based forecasts, the NDJF precipitation totals might compensate this monthly scale differences resulting in a good match of SWE values in February. This is obvious for many of the winter seasons shown in Fig. 5 (e.g., 1998/99 and 2000/01) and confirms the previous finding that improved model skill is possible when storages instead of instantaneous fluxes are considered.

### 3.3 The role of temperature and precipitation for SWE forecasts

In order to show the importance of both temperature and precipitation in SWE forecasting, Tab. 1 summarises the skill measure previously introduced for two other model experiments in which either temperature or precipitation is replaced by climatological forecasts: (i) Temperature from climatology is combined with precipitation forecasts from the climate models (second column of Tab. 1) and (ii) Precipitation from climatology is combined with temperature forecasts from the climate models (third column of Tab. 1). If one variable is replaced by climatology the standard deviation of anomalies is zero since the climatological forecasts have no deviations from climatology. This is in line with zero skill in terms of BSS and MAESS (see, temperature skills in the second column and precipitation skills in the third column, respectively). The skill measures of the respective variable that has not changed in this way is subject to the same characteristics as the full dynamical run (first column). For instance, if temperature is replaced by climatology, precipitation skills are equal to the full dynamic run (e.g., compare column one and two for precipitation).

In case of SWE, the effects of replacing either temperature or precipitation differ in terms of model skill. First, a drop in correlation is obvious in both cases. If temperature is replaced by climatology the hitrate of GloSea5-AWARE decreases only slightly to 10 but remains 7 in case of CFSv2-AWARE. If precipitation is replaced by climatology hitrates decrease in both

cases and the standard deviation is much lower than in the full dynamic run. This indicates that the variability in SWE forecasts is mainly prescribed by precipitation in the current study setup. However, the influence of temperature would likely increase for predictions of SWE in the ablation season.

Surprisingly, the RMSE in terms of SWE re-forecasts is lowest in the model run in which precipitation is replaced by climatology. Since this finding is neither confirmed by comparing MAESS (which computes similar error statistics but with linear instead of quadratic weighting of errors) values nor by considering any of the other skill measures, it is likely that this effect could be explained by the low variability of SWE in this experiment combined with the quadratic weighting of errors in RMSE computations. This comparison underlines the need for different skill measures in the process of evaluating forecasts.

## 3.4 Model skill and its relation to other studies

Compared to findings reported in the literature, the results achieved in this study are promising given that the skill in Europe is generally found to be low. For instance, according to Weisheimer and Palmer (2014) the skill of DJF temperature is "marginally useful" using ECMWF's System4. The rating for DJF precipitation is even found to be "not useful" (cf., Fig. 5 in Weisheimer and Palmer, 2014). Similarly, Kim et al. (2012) found some skill in terms of correlation for wintertime temperature predictions using System4. However, their study also suggests low absolute correlation coefficients for precipitation forecasts and for both temperature and precipitation forecasts achieved using CFSv2. A direct comparison to the results presented in this study is not possible since GloSea5 was not addressed in these studies. Moreover, given that only one single catchment is considered, a ranking of models is beyond the scope of this article. The predictability for SWE detected in this study can be related to both some skill in precipitation prediction and previous findings found for the persistence in SWE predictions for smaller forecast horizons. For instance, in case of the alpine snow cover, Jörg-Hess et al. (2015) underline the persistence in SWE predictions at least up to a lag of two weeks.

## 4 Conclusions

In this study, a systematic evaluation of CM-based seasonal winter forecasts starting in November has been performed using a water balance model. A new method has been developed focussing on hydrological storages instead of instantaneous hydrological fluxes. SWE was chosen as predictand here and two independent climate models were used as input data for monthly scale distributed water balance model. A robust approach based on standardised anomalies was applied in order to bridge the gap in scale between GCMs and the water balance model. In this way, basin-scale averages of temperature and precipitation depth are temporally integrated in order to achieve November to February (NDJF) averages and totals, respectively. Given a lead time of 4 months, the application of the water balance model then allows predicting SWE in February which is relevant for many sectors like water management or hydro power generation. Based on year-by-year evaluation of re-forecasts using different skill measures and a Binomial test, the results achieved using GloSea5-AWARE and CFSv2-AWARE indicate that dynamical (CM-based) seasonal forecasts can provide skill. A sensitivity analysis using different configurations of input datasets showed that SWE forecasts benefit from the skill in precipitation forecasts, especially in terms of variability and hitrate / Brier Score.

These findings might be related to the hydro-climatological characteristics of the study area where snow accumulation is the major process during winter while snowmelt as a strong temperature dependent process is less important in this time (Fig. 5). In other environments the relative role of temperature and precipitation might look different.

Regarding predictability, the location of the study area is also of particular interest in the process of interpreting the results. Due to the fact that the Alps are situated in a transition zone between northern and southern Europe, the influence of large-scale climate patterns, such as the North Atlantic Oscillation (NAO), should be analysed more detailed in the future. It is also known that El Niño Southern Oscillation (ENSO) impacts the climate in Europe in late winter and stratospheric sudden warmings are also important (Ineson and Scaife, 2008; Scaife et al., 2016). A first assessment of possible connections between the NAO on the one hand and snow and glacier related states on the other hand only resulted in low correlations (c.f. Beniston and Jungo, 2002; Scherrer et al., 2004; Bartolini et al., 2009; Marzeion and Nesje, 2012). However, in the southern and western parts of the Alps this relationship between NAO and snow and ice properties might be explained more clearly. Recent skill improvements regarding CM-based seasonal predictions might explain our detectable skill (Scaife et al., 2014). Future work should address climatological processes that are related to model skill and involve other basins in different parts of the Alps.

Besides studying the climatological perspective of predictability, the results also revealed uncertainties involved in hydrological modelling using the water balance model and scaling issues regarding the representativeness of point scale SWE observations. These findings also suggest improvements regarding both the provision of basin-scale SWE observations and the water balance model as an outlook for future work. Low flow conditions in March might be better predicted if the model would account for artificial reservoirs in the study area. Moreover, a better representation of changes in glaciated area is currently being investigated through coupling AWARE with a glacier evolution model developed by Marzeion et al. (2012). These features will be added to the model in the future.

However, the results of this study show that it is possible to detect skilful signals from dynamical (CM-based) seasonal predictions of hydrological storages in Europe, where seasonal predictions are still challenging. The results suggest that some seasonal predictions may be capable of predicting tendencies of hydrological model storages, although the skill of these predictions is in many cases low in Europe. The basic idea of this study is that a focus on hydrological storages rather than on hydrological fluxes might help in exploiting seasonal predictions. The first results of the methodology are promising with respect to practical applications in which hitrates above 70% might be seen as a reasonable target accuracy. Since snowmelt predictions are of particular interest in the study area, a similar approach could be applied to CM-based seasonal forecasts initialised in May. Future research should also address predictability studies in other regions. Moreover, it would be interesting to study the predictability of other hydrological storages like, e.g., glaciers, lakes or groundwater. A focus on probabilistic forecasts also is an interesting prospect for the future.

## 5  Data availability

– CFSv2 re-forecast data: https://nomads.ncdc.noaa.gov/data/cfsr-rfl-mmda/flxf/

– HISTALP: http://www.zamg.ac.at/histalp/index.php

## Appendix A: Model performance and skill measures

The definition of the Nash-Sutcliffe model efficiency $E$ and the benchmark Nash-Sutcliffe model efficiency $E_b$ reads (Schaefli and Gupta, 2007):

$$E_b = 1 - \frac{\sum_{t=1}^{N} \left[ q_{\text{obs}}(t) - q_{\text{sim}}(t) \right]^2}{\sum_{t=1}^{N} \left[ q_{\text{obs}}(t) - q_{\text{bench}}(t) \right]^2} \tag{A1}$$

In this computation time series of observed $q_{\text{obs}}$ and modelled $q_{\text{sim}}$ quantities are considered for all time steps $t$. $q_{\text{bench}}(t)$ is the time dependent benchmark value at timestep $t$. $q_{\text{bench}}(t)$ is a long-term average computed for the month of time step $t$. The original definition of Schaefli and Gupta (2007) refer to daily series for which the long-term average for a specific calendar day is applied. According to Schaefli and Gupta (2007) $E_b$ indicates if the model "has greater explanatory power than already contained in the seasonality of the driving forces (the climate)". If $q_{\text{bench}}(t) = \bar{q}_{\text{obs}}$ is assumed, $E_b$ is equal to the Nash-Sutcliffe

model efficiency $E$. In contrast to $E$, $E_b$ presumes the climatological mean of each time step as a benchmark against which all elements of the time series are compared. Since seasonality is inherent in many time series, generally $E_b \leq E$ holds.

A widely used measure to evaluate the accuracy of forecasts is the Brier Score $B$ (Wilks, 2006):

$$B = \frac{1}{N} \sum_{i=1}^{N} (f_i - o_i)^2 \tag{A2}$$

It is a special case of the Ranked Probability Score (RPS, see, e.g., Hersbach, 2000) that restricts the evaluation of forecasts

to two categories (e.g., above or below average). The forecast $f_i$ computed for each year $i$ is compared to the corresponding state observed in that year $o_i$, whereby $f_i$ and $o_i$ are dichotomous states (0 or 1). The Brier score $B$ is the average of the squared differences between $f_i$ and $o_i$. The average refers to a range of $N$ years. The best value that can be achieved in this way is zero indicating a perfect forecast skill. In contrast, 1 would indicate that all forecasts are wrong.

Another skill measure for forecasts is the Mean Absolute Error (MAE) which characterises, similar to RMSE, differences

between the forecasted value $q_f$ and the observed value $q_o$ (in units of the underlying time series):

$$M = \frac{1}{N} \sum_{i=1}^{N} |q_{f,i}(t) - q_{o,i}(t)| \tag{A3}$$

If $|q_{f,i}(t) - q_{o,i}(t)|$ is replaced by $(q_{f,i}(t) - q_{o,i}(t))^2$ and the square root is calculated from Eq. A3, this equation yields the Root Mean Square Error (RMSE). In contrast to the latter, MAE is less sensitive to larger differences between $q_{f,i}$ and $q_{o,i}$. Moreover, the MAE is comparable to the Continuous Ranked Probability Score (CRPS) used for probabilistic forecasts

(Hersbach, 2000; Trinh et al., 2013) and can be used for single-value (deterministic) forecasts.

In order to compare these skill measures computed for different forecasts to a reference forecast (i.e., climatology), a skill score $S$ measure is typically calculated. For instance, the MAE skill score ($S_{\mathrm{MAESS}}$) can be derived using

$$S_{\mathrm{MAESS}} = 1 - \frac{M_{\mathrm{forecast}}}{M_{\mathrm{reference}}}, \tag{A4}$$

with $M_{\mathrm{forecast}}$ is the MAE of the forecast system and $M_{\mathrm{reference}}$ is the MAE of the climatological forecast. Similarly, Eq. A4 can be applied to derive a Brier Skill Score $S_{\mathrm{BSS}}$ through replacing $M$ by $B$.

*Author contributions.* K. Förster prepared the manuscript with contributions from all co-authors, designed the study, performed the water balance simulations and predictability analyses. He and F. Hanzer developed the AWARE model which has been designed for this kind of study. E. Stoll contributed to downscaling of climate model output and reviewed the literature with respect to connections between snow and glacier observations on the one hand and the NAO index on the other hand. A. A. Scaife and C. MacLachlan computed and provided the GloSea5 re-forecasts and helped with data usage, interpretation of the results and improving the methodology. Snow observations in the study area were evaluated by J. Schöber who also contributed to interpreting anomalies in SWE. M. Huttenlau coordinated the project. S. Achleitner and U. Strasser are key researchers of the project. They supervised the scientific work and helped discussing the results and improving the methodology.

*Competing interests.* The authors declare that they have no conflict of interest.

*Acknowledgements.* This work was carried out as part of the project "W01 MUSICALS II – Multiscale Snow/Ice Melt Discharge Simulation for Alpine Reservoirs" at alpS – Centre for Climate Change Adaptation in Innsbruck, Austria. The K1-Centre alpS is funded through the Federal Ministry of Transport, Innovation and Technology (BMVIT), the Federal Ministry of Science, Research and Economy (BMWFW), as well as the Austrian federal states of Tyrol and Vorarlberg within the scope of COMET – Competence Centers for Excellent Technologies. The Programme COMET is managed by the Austrian Research Promotion Agency (FFG). We want to thank TIWAG – Tiroler Wasserkraft AG for the collaboration and co-funding the project. Another thanks goes to the NOAA (National Oceanic and Atmospheric Administration) National Centers for Environmental Prediction (NCEP) for the provision the CFSv2 data. The retrospective forecasts of the GloSea5 model were kindly provided by SPECS project ("Seasonal-to-decadal climate Prediction for the improvement of European Climate Services", http://www.specs-fp7.eu/). We would like to thank Felix Oesterle who wrote the script to automatically retrieve and convert CFSv2 data. Assistance with HISTALP data provided by Anna-Maria Tilg and Barbara Chimani is greatly appreciated. A.A.S. and C.M. were supported by the joint DECC/Defra MetOffice Hadley Centre Programme (GA01101). The publication of this article was funded by the Open Access fund of Leibniz Universität Hannover. We wish to thank two anonymous reviewers for their helpful comments that helped to improve the manuscript.

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

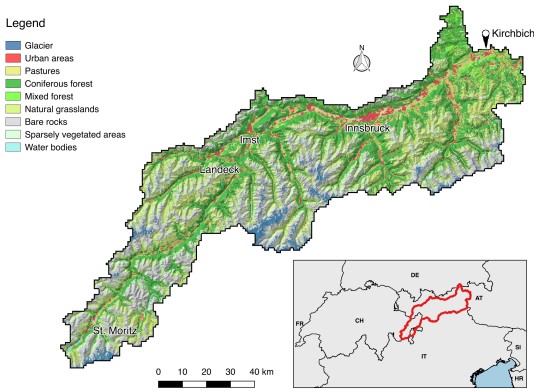

**Figure 1.** Map of the Inn headwaters upstream of the Kirchbichl gauging station. Major land use classes are shown along with the topography.

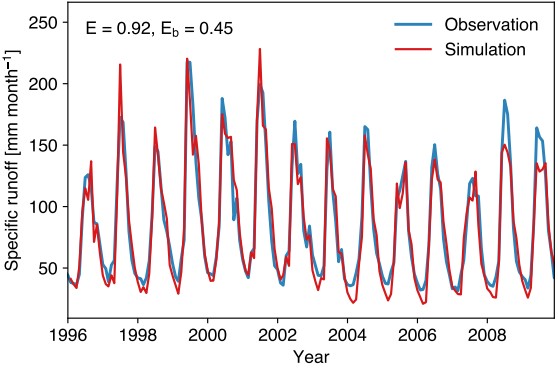

**Figure 2.** Specific runoff hydrographs for observed and modelled runoff at Kirchbichl gauging station and the corresponding performance measures $E$ (Nash-Sutcliffe model efficiency, NSE) and $E_b$ (benchmark NSE).

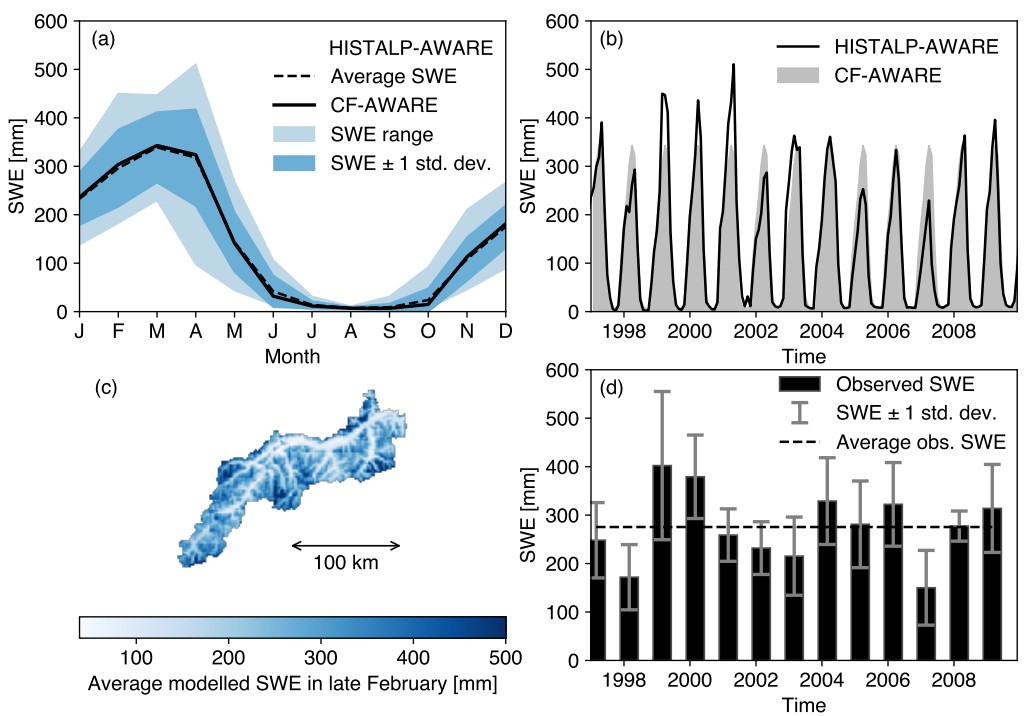

**Figure 3.** Observed and simulated snow states. (a) Range and standard deviation of SWE time series computed for single years and the corresponding mean monthly averages derived from AWARE using HISTALP data (HISTALP-AWARE). Average SWE conditions computed using climatological forecasts (CF-AWARE) are displayed as well. (b) This plot shows the same data but as time series between 1997-2009. (c) Map showing the average spatial SWE distribution computed for February. (d) SWE observations in February collected from stations in the study area above 1400 m a.s.l. (Schöber et al., 2016). The error bars show the ± 1 standard deviation of observations for each year (see Schöber et al., 2016, for further details on SWE sampling).

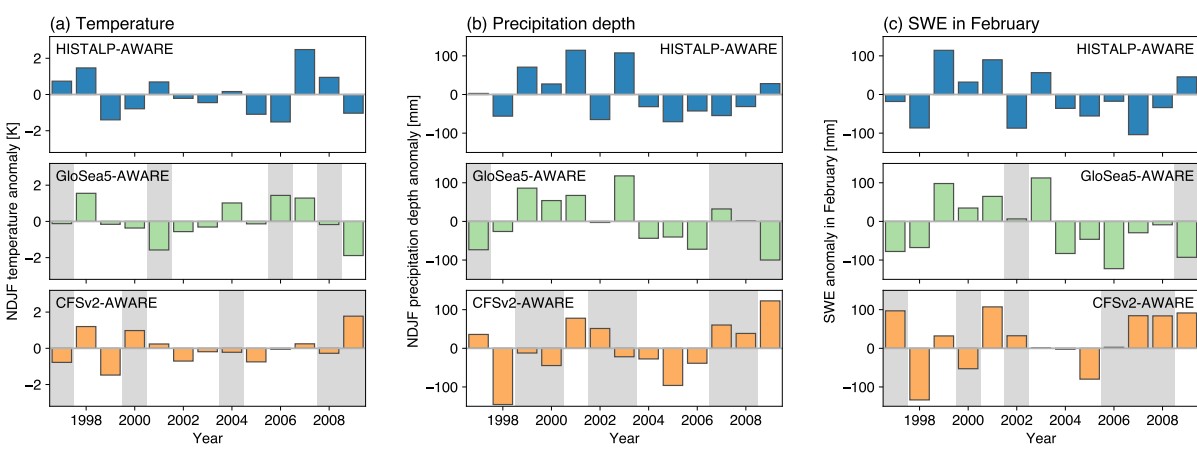

**Figure 4.** Anomalies of (a) basin-scale NDJF temperature, (b) NDJF areal precipitation depth, and (c) snow accumulation in February. Water balance simulations driven by CM-based seasonal forecasts are compared to water balance simulation driven by HISTALP. Shaded areas indicate years in which the sign of anomalies does not match the reference run.

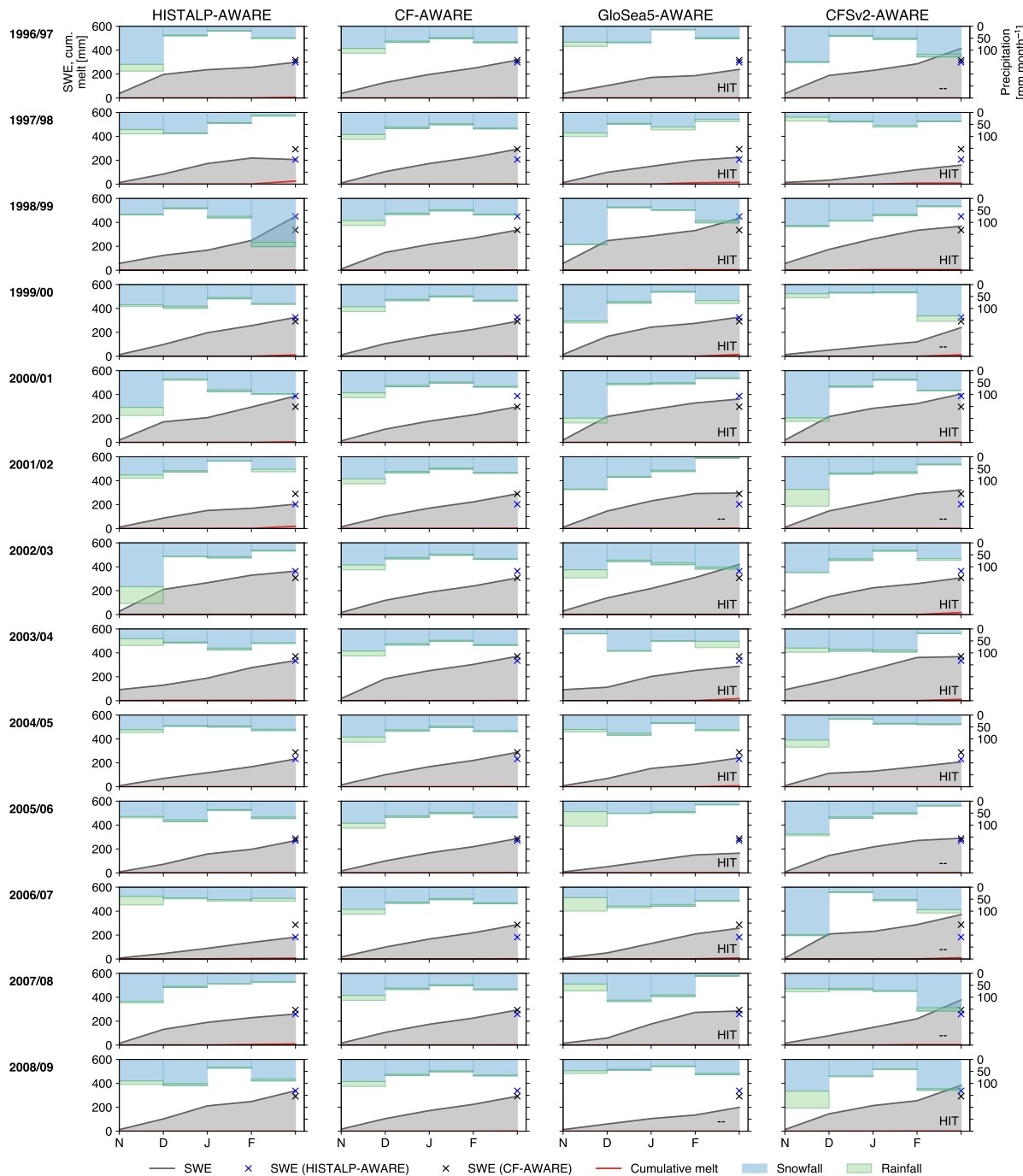

**Figure 5.** Water balance of the basin-scale snow storage for each year and each forcing dataset used for AWARE simulations. CF is the climatological forecast (longterm averages of HISTALP) which can be viewed as forecast yielding average conditions. The evolution of snow accumulation is categorized either "HIT" or "–" if the sign of anomalies obtained from HISTALP-AWARE and CM-based AWARE runs matches or mismatches, respectively.

**Table 1.** Skill measures for basin-scale averages of NDJF temperature, NDJF precipitation, and SWE in February. For each experiment, results for GloSea5-AWARE and CFSv2-AWARE are summarised. The first experiment (full dynamical model runs) refers to the standard setting of CM-based seasonal forecasts using AWARE (Sect. 3.2), while the other two experiments also involve a replacementof CM-based time series by climatology (Sect. 3.3). Skill measures: $r$: Pearson correlation coefficient, Std.: standard deviation of time series, RMSE: Root Mean Square Error, Hitrate: number of correctly predicted states (out of 13), BSS: Brier Skill Score, MAESS: Mean Absolute Error Skill Score. Please note: The units of Std. and RMSE are those of the input time series. HISTALP-AWARE is the reference used in the calculations of $r$, RMSE, Hitrate, BSS, and MAESS.

| | Full dynamical model runs | | Temperature from climatology | | Precipitation from climatology | |
|---|---|---|---|---|---|---|
| | GloSea5-AWARE | CFSv2-AWARE | GloSea5-AWARE | CFSv2-AWARE | GloSea5-AWARE | CFSv2-AWARE |
| **Temperature** | | | | | | |
| r [-] | 0.32 | 0.17 | - | - | 0.32 | 0.17 |
| Std. [K] | 1.03 | 0.86 | 0.00 | 0.00 | 1.03 | 0.86 |
| RMSE [K] | 1.29 | 1.32 | 1.16 | 1.16 | 1.29 | 1.32 |
| Hitrate [-] | 9 | 8 | - | - | 9 | 8 |
| BSS [-] | 0.69 | 0.62 | 0.00 | 0.00 | 0.69 | 0.62 |
| MAESS [-] | -0.02 | -0.12 | 0.00 | 0.00 | -0.02 | -0.12 |
| **Precipitation** | | | | | | |
| r [-] | 0.61 | 0.31 | 0.61 | 0.31 | - | - |
| Std. [mm] | 64.77 | 70.95 | 64.77 | 70.95 | 0.00 | 0.00 |
| RMSE [mm] | 55.97 | 78.66 | 55.97 | 78.66 | 62.20 | 62.20 |
| Hitrate [-] | 9 | 7 | 9 | 7 | - | - |
| BSS [-] | 0.69 | 0.54 | 0.69 | 0.54 | 0.00 | 0.00 |
| MAESS [-] | 0.19 | -0.29 | 0.19 | -0.29 | 0.00 | 0.00 |
| **SWE** | | | | | | |
| r [-] | 0.57 | 0.28 | 0.44 | 0.25 | 0.51 | 0.16 |
| Std. [mm] | 72.57 | 71.91 | 62.69 | 68.03 | 23.01 | 20.32 |
| RMSE [mm] | 65.27 | 88.02 | 68.88 | 90.03 | 59.32 | 66.99 |
| Hitrate [-] | 11 | 7 | 10 | 7 | 8 | 6 |
| BSS [-] | 0.85 | 0.54 | 0.77 | 0.54 | 0.62 | 0.46 |
| MAESS [-] | 0.14 | -0.23 | 0.06 | -0.19 | 0.13 | -0.05 |