# Peer review of "Retrospective forecasts of the upcoming winter season snow accumulation in the Inn headwaters (European Alps)"

_Hydrology and Earth System Sciences, 2017_

## Referee Comment (RC1) · Anonymous Referee #1 · 19 Jul 2017

**Summary**

In this paper the authors analyze seasonal hydrological hindcasts following a dynamical approach. The hydrological model AWARE is forced with the output from two different GCMs (CFSv2, GloSea5) for the Alpine catchment Inn up to the gauge Kirchbichl. As main predictand the authors chose the snow accumulation, represented by the snow water equivalent, at the end of February with a lead-time of 4 months (forecast initialization in October the preceding year). Additionally this paper assesses the predictive skill of both GCM-based forecast with regard to the anomalies of basin-scale mean temperature and accumulated precipitation depth.

In my opinion, the manuscript fits well into this special issue and its content is relevant for publication in HESS. In particular I like the use of two different GCMs in combination with a water balance model and the analyses of a different hydrological predictand than flow. I recommend this paper to be published after the authors have addressed the following general and specific comments in order to further improve the manuscript.

General comments

Overall, the paper is well written and well organized presenting interesting results. Nevertheless the authors should elaborate the following aspects: To facilitate readability of the manuscript you should explicitly indicate throughout the paper, if you're talking about meteorological seasonal forecast (e.g. used as input for a water balance model) or hydrological seasonal forecasts (output from a water balance model), as both cases are relevant in different parts of the text. I recommend to use the notation of the 4 model experiments (introduced in section 2.4) more consistently throughout the whole paper (including figures), for example clim. forecast should be CF-AWARE.

I suggest re-arranging the validation of the hydrological model: in chapter 2.3 you solely assess the performance of the water balance model with regard to runoff, although the prediction of SWE is the focal point of this study. Therefore I suggest moving the SWE-related evaluation from section 3 (page 7) to section 2.3 and to focus primarily on SWE simulation.

Although you focus on climate model-based seasonal forecast you should add some more background information on the different approaches to create seasonal hydrological forecast (e.g. statistical methods vs. dynamical approaches) in the introduction. Furthermore I miss a better support of literature in the discussion (section 3), e.g. with regard to related studies of the Alpine region, e.g. Fundel, F., Jörg-Hess, S., and Zappa, M.: Monthly hydrometeorological ensemble prediction of streamflow droughts and corresponding drought indices. Hydrol. Earth Syst. Sci., 395-407, doi:10.5194/hess-17-395-2013, 2013 and with regard to the comparison of your skill

measures (e.g. correlation) with other studies, e.g. Kim, H.M. , Webster, P. J., Curry, J. A.: Seasonal prediction skill of ECMWF System 4 and NCEP CFSv2 retrospective forecast for the Northern Hemisphere Winter. Clim Dyn (2012) 39:2957–2973. doi 10.1007/s00382-012-1364-6 or Weisheimer, A., and Palmer, T.N.: On the reliability of seasonal climate forecasts. Journal of The Royal Society Interface 11, Heft 96, S. 20131162–20131162, 2014.

You focused on "hydrological storages instead of instantaneous hydrological fluxes", which you call "a new aspect". For me it is not obvious . . . . . . if you already tried predicting fluxes for the melting season in the Alps? Is SWE indeed better predictable than the resulting flow? and . . . if SWE is a useful information / predictor for the stakeholder you mention (reservoir managers)? Is it as useful as flow forecasts or is it more like a fallback option?

As you evaluate forecast skill for SWE together with areal precipitation and temperature, I think it's necessary to address the interaction of these parameters more explicitly: what is the relative contribution of precipitation compared to temperature on SWE in the Alpine basin at the end of February? I guess you have insights into this interaction, so please share it with the reader of your paper.

Specific comments

I attach the specific comments as supplement.

Please also note the supplement to this comment:
https://www.hydrol-earth-syst-sci-discuss.net/hess-2017-370/hess-2017-370-RC1-supplement.pdf

**Supplement:**

**Specific comments: "Retrospective forecasts of the upcoming winter season snow accumulation in the Inn headwaters (European Alps)" by Kristian Förster et al.**

Page 1, line 13: I agree that seasonal forecasting is a topic which became a focal point in hydrological forecasting in recent years. But the term "new" isn't adequate in my opinion, as the earliest references e.g. for ESP-based long-term forecasts have been published in the 1970s / 1980s (e.g. Day, G. (1985). "Extended Streamflow Forecasting Using NWSRFS." *J. Wat. Res. Plan. Mgmt.*, 10.1061/(ASCE)0733-9496(1985)111:2(157), 157-170 or Twedt, T. M., J. C. Schaake, Jr., and E. L. Peck (1977). National Weather Service extended streamflow prediction. *Proc., Western Snow Conference*, 52 – 57.). Furthermore on page 2, line 8 you state that "seasonal outlooks […] have been prepared for decades." – This is a contradiction to the term "new", too.

Page 2, line 12/13: This sentence is a bit confusing / misleading, because meteorological data is used for the ESP-approach, too. Furthermore I recommend extending your description of ESP / revESP in order to explain more clearly which components contribute to forecast skill in each approach.

Page 2, line 18: I suggest adding "on the one hand", because otherwise the sentence might be misleading as seasonal forecast aren't solely an initial state problem (as you mention below).

Page 3, line 9: Please be more precise by adding "…based on seasonal predictions in the Alps" (or something similar).

Page 3, line 15: I suggest adding that the Inn basin belongs to the catchment of the Danube and that the Inn is the main tributary of the Upper Danube.

Page 4, line 5: Which "multi-year period" did you chose (the whole HISTALP period)?

Page 4, line 6: Please add a reference and explain what you mean by "randomly selecting valid values".

Page 4, line 13-26: Please add the temporal and spatial resolution of the output from both GCMs you're using in your model.

Page 4, line 24: In line 28 you state, that only re-forecasts starting in November are considered, but here also "25 Oct." is listed as initial start date. Please explain.

Page 4, line 31: What is the grid-size of the AWARE model used in this study?

Page 5, 7-9: I suggest splitting this sentence in order to make it easier to read.

Page 5, line 27: Do you recognize mismatches in summer / autumn, when the reservoirs are filled-up, too? Please comment on this.

Page 6, line 17-20: I suggest adding the number of ensemble-members for each AWARE-run. This will help the reader remembering the set-up you described in section 2.2.3

Page 7, line 16: I don't see the information / added value of Fig. 3 (c) for the reader, because there's no comparison to measured SWE. Please explain why you decided to include this figure.

Figure 3 (d): You should add the errors bars to the legend.

Page 8, line 8: As far as I understand, the hit rate for precipitation is equal (GloSea5) or lower (CFSv2) compared to temperature and higher as you state.

Page 8, line 8: As your finding that "The skill in precipitation predictions is higher [than temperature]" isn't something I would have expected before, you should extent discussing this result more detailed and refer to similar and contradicting results from other studies.

Page 8, line 11-13: You state that predicting of hydrological storages (SWE in your case) is more robust / skillful than predicting fluxes; You should prove this statement. As your hydrological model also generates flows, I suggest relating this statement to your model results.

Page 8, line 26-28: Without doubt the representativeness of measured SWE and its interpolated on basin scale is problematic. But in my opinion you should mention, that the water balance model is never perfect and that it introduces uncertainties into hydrological forecasts, too. I guess it is your intention to exclude (at this state) the hydrological model-related errors by using a reference run?

Page 8, line 28-29: I suggest to use "GCM-forecast skill" (or something similar) instead of "model skill when using CM-based forecasts".

Page 8, line 31: The cumulative snow melt is very difficult to recognize in figure 5 as its value is very small. So why did you plot this parameter (you don't use this information in the text anymore)? I think it can be skipped.

Page 9, line 2: Please add "… and CFSv2-AWARE".

Page 9, line 10: Please mention, which aspects of your method are really "new" (e.g. predictand, …).

Page 10, line 3: Interesting statement. Could you please comment on the definition of the "target accuracy". Who defined this Was is defined by users / stakeholders?

---

## Referee Comment (RC2) · Anonymous Referee #2 · 2 Aug 2017

This study explores the forecast skill of snow water equivalent (SWE) by using CGCM-driven water balance model simulations over a headwater region. While the topic is quite interesting and some results (e.g., GloSea5-driven forecasts) are potentially promising, the manuscript could be further improved after addressing several comments below.

Major comments:

1. An interesting question that could be answered in this manuscript is whether precipitation or temperature prediction more important for the SWE forecasting over the headwater region. Although precipitation prediction is less skillful than temperature in

many cases, the study region shows less skillful temperature prediction, perhaps due to the deficiency in snow or frozen soil processes. To compare their relative roles, precipitation or temperature forecast could be replaced with climatology before driving the SWE model. Such comparison would provide implications in advancing SWE forecasting.

2. The study shows that GloSea5-driven SWE forecasting is better than the CFSv2-driven forecasting in terms of pearson correlation for the ensemble mean, but did not tell why the former is better? Some information on precipitation and temperature forecasts could be mentioned in the abstract. Moreover, probabilistic metrics (e.g., RPSS) is needed besides just simply using correlation. Given that this manuscript is not a short communication, I would encourage the authors to have a more comprehensive evaluation for SWE forecasting.

Minor comments:

3. Does the AWARE water balance model distinguish the input of liquid or solid precipitation? If so, how to obtain the solid precipitation from global climate forecast model like CFSv2?

4. What is spatial resolution for the AWARE model over the study catchment?

5. What is the definition for the benchmark Nash-Sutcliffe efficiency?

6. For the benchmark NSE during the validation period, why does it drop to 0.25? Is it because there is trend or non-stationarity in the time series?

7. Figure 4. Besides correlation, how about the RMSE for the prediction?

8. Figure 4. Is the model-simulated SWE or observed SWE used for verification? If the former, how to demonstrate the usefulness of the SWE forecasting given the limited skill in SWE simulation with AWARE (where NSE=0.25 in the validation period)?
* * *
370, 2017.

---

## Author Comment (AC1) · 16 Oct 2017

**Reply to anonymous reviewer #1**

*(Reviewer's comments are in italics)*

*Summary*

*In this paper the authors analyze seasonal hydrological hindcasts following a dynamical approach. The hydrological model AWARE is forced with the output from two different GCMs (CFSv2, GloSea5) for the Alpine catchment Inn up to the gauge Kirchbichl. As main predictand the authors chose the*

[Figure]

*snow accumulation, represented by the snow water equivalent, at the end of February with a lead-time of 4 months (forecast initialization in October the preceding year). Additionally this paper assesses the predictive skill of both GCM-based forecast with regard to the anomalies of basin-scale mean temperature and accumulated precipitation depth.*

*In my opinion, the manuscript fits well into this special issue and its content is relevant for publication in HESS. In particular I like the use of two different GCMs in combination with a water balance model and the analyses of a different hydrological predictand than flow. I recommend this paper to be published after the authors have addressed the following general and specific comments in order to further improve the manuscript.*

We would like to thank anonymous referee #1 for his/her positive and constructive review of our discussion paper. Your comments and suggestions will help us in the process of revising our manuscript. Please find our detailed response below.

*General comments*

*Overall, the paper is well written and well organized presenting interesting results. Nevertheless the authors should elaborate the following aspects: To facilitate readability of the manuscript you should explicitly indicate throughout the paper, if you're talking about meteorological seasonal forecast (e.g. used as input for a water balance model) or hydrological seasonal forecasts (output from a water balance model), as both cases are relevant in different parts of the text. I recommend to use the notation of the 4 model experiments (introduced in section 2.4) more consistently throughout the whole paper (including figures), for example clim. forecast should be CF-AWARE.*

Yes, this is a good point. We carefully updatde the manuscript with respect to the

notation of model experiments. We are expecting to increase the readability in this way. Thank you for this recommendation.

> *I suggest re-arranging the validation of the hydrological model: in chapter 2.3 you solely assess the performance of the water balance model with regard to runoff, although the prediction of SWE is the focal point of this study. Therefore I suggest moving the SWE- related evaluation from section 3 (page 7) to section 2.3 and to focus primarily on SWE simulation.*

We discussed moving the SWE evaluation to the model description section. At first sight, it would be a good alternative which seems to be straight-forward. However, this would increase the number of cross references in the manuscript, which, in our opinion, might not be helpful in the process of revising our manuscript. The model is calibrated and validated using runoff. This makes sense since the scales of the model and the observations match. We didn't calibrate against SWE data. Moreover, the evaluation of SWE also includes a study of the representativeness of the climatological forecast (CF-AWARE). This type of forecasts is introduced in the section describing the model experiment (2.4) which follows the model description section (2.4). Moreover, it also depends on initial conditions. Therefore, we see the SWE comparison as part of the results section.

However, we agree that we should highlight the relevance for presenting these results here. We will add a piece of introducing text prior to this analysis:

"While the applicability of AWARE to reconstruct the water balance in terms of observed runoff time series was demonstrated in Sect 2.3, it is necessary to evaluate the model experiments HISTALP-AWARE and CF-AWARE with respect to SWE prior to the analyses of CM-based SWE forecasts."

> *Although you focus on climate model-based seasonal forecast you should*

*add some more background information on the different approaches to create seasonal hydrological forecast (e.g. statistical methods vs. dynamical approaches) in the introduction. Furthermore I miss a better support of literature in the discussion (section 3), e.g. with regard to related studies of the Alpine region, e.g. Fundel, F., JoÌLrg-Hess, S., and Zappa, M.: Monthly hydrometeorological ensemble prediction of streamflow droughts and corresponding drought indices. Hydrol. Earth Syst. Sci., 395-407, doi:10.5194/hess-17-395-2013, 2013 and with regard to the comparison of your skill measures (e.g. correlation) with other studies, e.g. Kim, H.M. , Webster, P. J., Curry, J. A.: Seasonal prediction skill of ECMWF System 4 and NCEP CFSv2 retrospective forecast for the Northern Hemisphere Winter. Clim Dyn (2012) 39:2957–2973. doi 10.1007/s00382-012-1364-6 or Weisheimer, A., and Palmer, T.N.: On the reliability of seasonal climate forecasts. Journal of The Royal Society Interface 11, Heft 96, S. 20131162–20131162, 2014.*

The revised version of the manuscript will include some more information about statistical and dynamical approaches in the introduction (please refer to our response to your specific comment on this topic below). In this context, we will also refer to the literature you have listed in your comment when discussing the results later in Section 3:

"Compared to findings reported in the literature, the results achieved in this study are promising given that the skill in Europe is generally found to be low. For instance, according to Weisheimer and Palmer (2014) the skill of DJF temperature is "marginally useful" using ECMWF's System4. The rating for DJF precipitation is even found to be "not useful" (cf., Fig. 5 in Weisheimer and Palmer, 2014). Similarly, Kim et al. (2012) found some skill in terms of correlation for wintertime temperature predictions using System4. However, their study also suggests low absolute correlation coefficients for precipitation forecasts and for both temperature and precipitation forecasts achieved using CFSv2. A direct comparison to the results presented in this study is not possible since GloSea5 was not addressed in these studies. Moreover, given that only one single catchment is considered, a ranking of models is beyond the scope of this article. The predictability for SWE detected in this study can be related to both some skill in precipitation prediction and previous findings found for the persistence in SWE predictions for smaller forecast horizons. For instance, in case of the alpine snow cover, Jörg-Hess et al. (2015) underline the persistence in SWE predictions at least up to a lag of two weeks."

> *You focused on "hydrological storages instead of instantaneous hydrological fluxes", which you call "a new aspect". For me it is not obvious . . . . . . if you already tried predicting fluxes for the melting season in the Alps? Is SWE indeed better predictable than the resulting flow? and . . . if SWE is a useful information / predictor for the stakeholder you mention (reservoir managers)? Is it as useful as flow forecasts or is it more like a fallback option?*

We agree that snowmelt predictions are very useful for numerous stakeholders. Accordingly we added additional literature to the introduction. Yet, we didn't try to predict fluxes using CM-based meteorological forcing for the melting season because we expect the predictions to be less accurate than for SWE in winter. We added the following lines:

"We focus on the winter season as extratropical seasonal forecasts appear to have the highest skill in this season (e.g. Riddle et al 2013, Scaife et al 2014, Kang et al 2014). There are a number of reasons for this, including winter being the season when the stratosphere is active, which is known to affect predictions (e.g. Domeisen et al 2015, Scaife et al 2016, Butler et al 2016). The winter season also shows much stronger dynamical connections to the tropics, allowing high predictability of tropical rainfall (Kumar 2013) to be transmitted into the extratropics (Greatbatch et al 2012, Molteni et al 2015, Scaife et al 2017)."

In a different model experiment, spring forecasts have been studied based on initial conditions derived from CM-based seasonal model runs and HISTALP meteorological forcing (Förster et al., 2017). However, CM-based seasonal forecasts have only been addressed to study snow accumulation in the winter and simulations in spring are performed in the framework of a reverse-ESP experiment. Thus, fully dynamical seasonal forecasts of spring runoff have not been performed yet. However, in the discussion paper, we already addressed snowmelt forecasts in spring as an outlook for future research in the conclusions section (page 10, line 3-4). This topic could be addressed in the framework of a subsequent analysis.

For the hydropower stakeholders peak SWE, which is the starting point of the melt season, would be beneficial for reservoir inflow and flood forecasting. In the Alps peak SWE normally occurs in April or May. In a first step, we tried to make use of wintertime forecasts and the predicted SWE in February is viewed as a first guess for the catchment's state prior to snowmelt. Moreover, SWE is important variable for tourism in the Alps. This argumentation was also added to the introduction.

Please refer to our reply to your specific comment on the question whether predicting storages is actually more robust instead of predicting fluxes.

> *As you evaluate forecast skill for SWE together with areal precipitation and temperature, I think it's necessary to address the interaction of these parameters more explicitly: what is the relative contribution of precipitation compared to temperature on SWE in the Alpine basin at the end of February? I guess you have insights into this interaction, so please share it with the reader of your paper.*

Evaluating the relative role of both temperature and precipitation in SWE forecasting is a question which has also been raised by reviewer #2. We will prepare a small model experiment in which climate model forcing is replaced by either climatological series

of temperature or precipitation, respectively. This experiment gains insight into the question whether temperature or precipitation mainly contributes to the skill in SWE forecasts. The revised version of the manuscript will include a table of performance measures in order to highlight the contribution of temperature and precipitation, respectively. Please also refer to our response to the comments of Anonymous Reviewer #2.

*Specific comments*

*I attach the specific comments as supplement.*

*Page 1, line 13: I agree that seasonal forecasting is a topic which became a focal point in hydrological forecasting in recent years. But the term "new" isn't adequate in my opinion, as the earliest references e.g. for ESP-based long-term forecasts have been published in the 1970s / 1980s (e.g. Day, G. (1985). "Extended Streamflow Forecasting Using NWSRFS." J. Wat. Res. Plan. Mgmt., 10.1061/(ASCE)0733-9496(1985)111:2(157), 157-170 or Twedt, T. M., J. C. Schaake, Jr., and E. L. Peck (1977). National Weather Service extended streamflow prediction. Proc., Western Snow Conference, 52 – 57.). Furthermore on page 2, line 8 you state that "seasonal outlooks [. . . ] have been prepared for decades." – This is a contradiction to the term "new", too.*

We agree that this argumentation in its present form is not consistent. Indeed, seasonal hydrological forecasting is a topic that is not so new as mentioned. Some additional literature was added to the introduction. However, using climate model based hydrological forecasting is new. Thus, we will revise the text regarding the historical perspective of its relevance. In the revised version, we will restrict the "new" aspect to climate model based seasonal predictions:

"Climate model (CM)-based seasonal predictions are an emerging new field in hydrology (e.g., Yuan et al., 2015; Svensson et al.,2015; Mackay et al., 2015)"

We will also add the references Day (1985) and Twedt et al. (1977) along with some other references, see below. Thank you for this important comment.

> *Page 2, line 12/13: This sentence is a bit confusing / misleading, because meteorological data is used for the ESP-approach, too. Furthermore I recommend extending your description of ESP / revESP in order to explain more clearly which components contribute to forecast skill in each approach.*

You are right to say that this statement is misleading. We extended our explanations regarding ESP/reverse-ESP and added the references as suggested:

"This methodology is well known and referred to as Ensemble Streamflow Prediction (ESP, Wood and Lettenmaier, 2008). The development of this method goes back to the seventies and eighties (Twedt et al., 1977; Day, 1985) and framed the development of statistical seasonal hydrological forecasting. ESP is a very useful method to study the influence of meteorological boundary conditions, which are obtained from observed long-term records, on the results of the hydrological forecasting model. In contrast, the reversed-ESP experiment is based on actual meteorological forcing but involves an ensemble of initial states, which makes it an appropriate method to study the influence of initial conditions on forecast results. The combination of both methods is also subject to recent research on predictability of hydrological systems (e.g. the VESPA approach, Wood et al., 2016)."

> *Page 2, line 18: I suggest adding "on the one hand", because otherwise the sentence might be misleading as seasonal forecast aren't solely an initial state problem (as you mention below).*

Done.

*Page 3, line 9: Please be more precise by adding "...based on seasonal predictions in the Alps" (or something similar).*

Done.

*Page 3, line 15: I suggest adding that the Inn basin belongs to the catchment of the Danube and that the Inn is the main tributary of the Upper Danube.*

Good point. We added some additional text.

*Page 4, line 5: Which "multi-year period" did you chose (the whole HISTALP period)?*

We added "(i.e., 1996-2009)".

*Page 4, line 6: Please add a reference and explain what you mean by "randomly selecting valid values".*

We dropped this sentence.

*Page 4, line 13-26: Please add the temporal and spatial resolution of the output from both GCMs you're using in your model.*
The spatial resolution is already mentioned in the brief descriptions of the climate models. However, we will add a further remark that confirms the use of the original grid spacing for water balance simulations. Now both spatial and temporal resolution are described clearly.

"Monthly grids of the climate models with their original grid spacing (as specified above) are used as forcing data for the water balance model which is described in the next section."

*Page 4, line 24: In line 28 you state, that only re-forecasts starting in November are considered, but here also "25 Oct." is listed as initial start date. Please explain.*

The run initialised on 25 Oct is part of the lagged ensemble as the first (complete) month in the output is November. We will add the fact that a lagged ensemble is applied.

*Page 4, line 31: What is the grid-size of the AWARE model used in this study? Page 5, 7-9: I suggest splitting this sentence in order to make it easier to read.*

The fact that the spatial resolution of the AWARE model was missing in the discussion paper was also addressed by the comments of reviewer #2 We are sorry that this information was missing. The spatial resolution of the model setup for the Inn headwaters is 1000 m. The revised version of the manuscript will definitely include this information. Moreover, the sentence you have mentioned is split.

*Page 5, line 27: Do you recognize mismatches in summer / autumn, when the reservoirs are filled-up, too? Please comment on this.*

[Figure]

In summer and autumn, a mismatch is also expected due to reservoir operations. In contrast to spring, when precipitation is almost completely accumulated in form of snow and a recession line is obvious in the runoff hydrograph, a similar attribution is not possible in summer/autumn because the mismatch in reservoir operation might be obliterated by rainfall. In order to better address both effects explicitly, a better representation of reservoirs is required in the model. We already mentioned improving the model description of reservoirs in the outlook (p. 9, lines 29-30).

*Page 6, line 17-20: I suggest adding the number of ensemble-members for each AWARE-run. This will help the reader remembering the set-up you described in section 2.2.3*

The total number of members of each CM ensemble is now added to the list of runs.

*Page 7, line 16: I don't see the information / added value of Fig. 3 (c) for the reader, because there's no comparison to measured SWE. Please explain why you decided to include this figure.*

In the discussion paper, Fig. 3c) is referred when averaging of snow conditions in February are explained. We agree that this context does not necessarily require additional material (e.g., a figure) to support these explanations. However, the figure shows the variability of SWE in February. We now refer to the figure in this context.

*Figure 3 (d): You should add the errors bars to the legend.*

Done.

*Page 8, line 8: As far as I understand, the hit rate for precipitation is equal (GloSea5) or lower (CFSv2) compared to temperature and higher as you state.*

We are sorry for these misleading explanations. We rewrote the next sentence indicating that correlation is higher in case of precipitation only. Please refer to the next comment.

> *Page 8, line 8: As your finding that "The skill in precipitation predictions is higher [than temperature]" isn't something I would have expected before, you should extent discussing this result more detailed and refer to similar and contradicting results from other studies.*

Regarding precipitation, it is well known that this is generally less skilfully predicted than temperature in most regions. We also acknowledge that while the results shown here indicate some skill, they do not show significantly different skill between temperature and rainfall and so this text has been altered:

"In the case of GloSea5-AWARE, the hitrate of correctly predicted anomalies regarding precipitation is [. . . ]"

> *Page 8, line 11-13: You state that predicting of hydrological storages (SWE in your case) is more robust / skillful than predicting fluxes; You should prove this statement. As your hydrological model also generates flows, I suggest relating this statement to your model results.*

You are right! The paper would really benefit if we would provide a quantitative assessment which proves this statement. However, we think that it is not possible to simply compare the accuracy of storages to corresponding values of runoff. In winter runoff is subject to low flow conditions affected by reservoir operation. Focussing on other seasons would make this comparison more difficult due to the different climatological and hydrological processes. Thus, we suggest focussing on precipitation instead because we can compare the range of monthly scale correlations to the corresponding NDJF values. We will add the following lines:

"In our study, we found monthly scale correlations computed for precipitation forecasts ranging from -0.29 to 0.30 (GloSea5-AWARE) and -0.11 to 0.15 (CFSv2-AWARE), respectively. These are generally lower than the corresponding values achieved for the averaged NDJF forecasts (GloSea5-AWARE: 0.61, CFSv2-AWARE: 0.31). Similar values of the same order have been observed for SWE forecasts (GloSea5-AWARE: 0.57, CFSv2-AWARE: 0.28)."

Thank you for pointing us in this direction!

> *Page 8, line 26-28: Without doubt the representativeness of measured SWE and its interpolated on basin scale is problematic. But in my opinion you should mention, that the water balance model is never perfect and that it introduces uncertainties into hydrological forecasts, too. I guess it is your intention to exclude (at this state) the hydrological model-related errors by using a reference run?*

We agree! Reviewer #2 also pointed us in this direction. At this stage, we exclude the hydrological model, which definitely introduces additional uncertainties. Instead, the comparison with the reference run (HISTALP-AWARE) is evaluated here. We rephrase this statement accordingly:

"However, the comparison between HISTALP-AWARE and the CM-based seasonal forecasts highlights GCM-forecast skill and acknowledges the fact that the water balance model is never perfect since it introduces uncertainties into hydrological forecasts, too. Due to the reasonably good agreement between seasonal forecasts and the reference run, the skill of CM-based forecasts is viewed promising."

> *Page 8, line 28-29: I suggest to use "GCM-forecast skill" (or something similar) instead of "model skill when using CM-based forecasts".*

Done. Please also refer to the previous comment.

> *Page 8, line 31: The cumulative snow melt is very difficult to recognize in figure 5 as its value is very small. So why did you plot this parameter (you don't use this information in the text anymore)? I think it can be skipped.*

You are right to say that snowmelt might be neglected in this chart. However, we think that this information is also important in this context since the water balance is closed. Otherwise the question about the relevance of snowmelt might arise.

> *Page 9, line 2: Please add "... and CFSv2-AWARE".*

Done. This information was also added in other parts of the text in order to make clear that these values represent basin averages derived using AWARE.

> *Page 9, line 10: Please mention, which aspects of your method are really "new" (e.g. predictand, ...).*

Indeed, the term "new" should be explained more detailed: We have added the following lines:

"SWE was chosen as predictand here and two independent climates model were used as input data for monthly scale distributed water balance model. A robust approach based on standardised anomalies was applied in order to bridge the gap in scale between GCMs and the water balance model."

> *Page 10, line 3: Interesting statement. Could you please comment on the definition of the "target accuracy". Who defined this Was is defined by users / stakeholders?*

The target accuracy is not a strictly defined threshold. The value should reflect an improvement over a 50:50 probability. Moreover, 70% is a realistic value of forecasts skill in the mid-latitudes which can be provided by climate models (Bell et al., 2017).

**References:**

Bell, V. A., Davies, H. N., Kay, A. L., Brookshaw, A., and Scaife, A. A.: A national-scale seasonal hydrological forecast system: development and evaluation over Britain, Hydrol. Earth Syst. Sci., 21, 4681–4691, doi:10.5194/hess-21-4681-2017, 2017.

Butler, A. H., Arribas, A., Athanassiadou, M., Baehr, J., Calvo, N., Charlton-Perez, A., Déqué, M., Domeisen, D. I. V., Frölhlich, K., Hendon,H., Imada, Y., Ishii, M., Iza, M., Karpechko, A. Y., Kumar, A., MacLachlan, C., Merryfield, W. J., Müller, W. A., O'Neill, A., Scaife, A. A., Scinocca, J., Sigmond, M., Stockdale, T. N., and Yasuda, T.: The Climate-system Historical Forecast Project: do stratosphere-resolving models make better seasonal climate predictions in boreal winter?, Quart. J. Roy. Meteor. Soc., 142, 1413–1427, doi:10.1002/qj.2743, 2016.

Day, G. N.: Extended Streamflow Forecasting Using NWSRFS, J. Water Resour. Plann. Manage., 111, 157–170, doi:10.1061/(ASCE)0733-9496(1985)111:2(157), 1985.

Domeisen D.I.V., Butler A.H., Fröhlich K., Bittner, M., Müller, W.A., Baehr, J.: Seasonal predictability over Europe arising from El Niño and stratospheric variability in the MPI-ESM seasonal prediction system. Journal of Climate 28: 256–271, doi: 10.1175/JCLI-D-14-00207.1, 2015.

Förster, K., Hanzer, F., Stoll, E., Schöber, J., Scaife, A. A., MacLachlan, C., Huttenlau, M., Achleitner, S., Strasser, U.: Probabilistic retrospective forecasts of snow accumulation for the upcoming winter season in the Inn headwaters catchment (Austria). Geophysical Research Abstracts, 19, EGU2017-15495, 2017.

Greatbatch, R.J., Gollan, G., Jung, T., Kunz, T.: Factors influencing Northern Hemi-

sphere winter mean atmospheric circulation anomalies during the period 1960/1961 to 2001/2002. Q. J. R. Meteorol. Soc. 138: 1970–1982, doi:10.1002/qj.1947, 2012.

Jörg-Hess, S., Griessinger, N., and Zappa, M.: Probabilistic Forecasts of Snow Water Equivalent and Runoff in Mountainous Areas, J. Hydrometeorol., 16, 2169–2186, 2015.

Kang, D., Lee, M.I., Im, J., Kim, D., Kim, H.-M., Kang, H.-S., Schubert, S.D., Arribas, A.A., MacLachlan, C.: Prediction of the Arctic Oscillation in boreal winter by dynamical seasonal forecasting systems. Geophysical Research Letters 10: 3577–3585, doi: 10.1002/2014GL060011, 2014.

Kim, H.-M., Webster, P. J., and Curry, J. A.: Seasonal prediction skill of ECMWF System 4 and NCEP CFSv2 retrospective forecast for the Northern Hemisphere Winter, Climate Dyn., 39, 2957–2973, doi:10.1007/s00382-012-1364-6, 2012.

Kumar, A., Chen, M., Wang, W.: Understanding prediction skill of seasonal mean precipitation over the Tropics. J. Clim. 26: 5674–5681, 2013.

Molteni, F., Stockdale, T.N., Vitart, F.: Understanding and modelling extratropical teleconnections with the Indo-Pacific region during the northern winter. Clim. Dyn. 45: 3119–3140, doi: 10.1007/s00382-015-2528-y, 2015.

Riddle, E.E., Butler, A.H., Furtado, J.C., Cohen, J.L., Kumar, A.: CFSv2 ensemble prediction of the wintertime Arctic Oscillation. Climate Dynamics 41: 1099–1116, 2013.

Scaife A.A. et al.: Tropical Rainfall, Rossby Waves and Regional Winter Climate Predictions. Quart. J. Roy. Met. Soc., DOI: 10.1002/qj.2910, 2017.

Scaife, A. A., Arribas, A., Blockley, E., Brookshaw, A., Clark, R. T., Dunstone, N., Eade, R., Fereday, D., Folland, C. K., Gordon, M., Hermanson, L., Knight, J. R., Lea, D. J., MacLachlan, C., Maidens, A., Martin, M., Peterson, A. K., Smith, D., Vellinga, M., Wallace, E., Waters, J., and Williams, A.: Skillful long-range prediction of European and North American winters, Geophys. Res. Lett., 41, 2514–2519, doi:10.1002/2014GL059637, 2014.

Scaife, A. A., Karpechko, A. Y., Baldwin, M. P., Brookshaw, A., Butler, A. H., Eade, R., Gordon, M., MacLachlan, C., Martin, N., Dunstone, N., and Smith, D.: Seasonal winter forecasts and the stratosphere, Atmos. Sci. Lett., 17, 51–56, doi:10.1002/asl.598, 2016.

Twedt, T. M., Schaake Jr, J. C., and Peck, E. L.: National Weather Service extended streamflow prediction, in: Proceedings Western Snow Conference, pp. 52–57, 1977.

Weisheimer, A. and Palmer, T. N.: On the reliability of seasonal climate forecasts, Journal of The Royal Society Interface, 11, 20131 162–20131 162, doi:10.1098/rsif.2013.1162, 2014.

Wood, A. W. and Lettenmaier, D. P.: An ensemble approach for attribution of hydrologic prediction uncertainty, Geophys. Res. Lett., 35, 15 L14 401, 5–5, doi:10.1029/2008GL034648, 2008.

---

## Author Comment (AC2) · 16 Oct 2017

**Reply to anonymous reviewer #2**

*(Reviewer's comments are in italics)*

> *This study explores the forecast skill of snow water equivalent (SWE) by using CGCM- driven water balance model simulations over a headwater region. While the topic is quite interesting and some results (e.g., GloSea5-driven forecasts) are potentially promising, the manuscript could be further improved after addressing several comments below.*

We would like to thank Anonymous Reviewer # 2 for his/her detailed review of our manuscript. The comments will help us in the process of improving the discussion paper.

*Major comments:*

*1. An interesting question that could be answered in this manuscript is whether precipitation or temperature prediction more important for the SWE forecasting over the headwater region. Although precipitation prediction is less skillful than temperature in many cases, the study region shows less skillful temperature prediction, perhaps due to the deficiency in snow or frozen soil processes. To compare their relative roles, precipitation or temperature forecast could be replaced with climatology before driving the SWE model. Such comparison would provide implications in advancing SWE forecasting.*

We appreciate your suggestion! Reviewer #1 also asked us to address the relative roles of temperature and precipitation more explicitly. Indeed, a model experiment that includes both dynamical and climatological forcing data helps to analyse the relevance of temperature and precipitation for SWE forecasts. We will follow your suggestion and we performed such a model experiment. Accordingly, the following runs are performed:

- temperature from models, precipitation from models (this is the configuration we have applied so far)
- temperature from models, precipitation from climatology
- temperature from climatology, precipitation from models

The second and the third model runs is analysed in the same way as already done with respect to the first model experiment. At this stage, we will also keep in mind your

suggestions outlined in the second comment of your review. All performance and skill measures will be summarised in an additional table.

*2. The study shows that GloSea5-driven SWE forecasting is better than the CFSv2- driven forecasting in terms of pearson correlation for the ensemble mean, but did not tell why the former is better? Some information on precipitation and temperature forecasts could be mentioned in the abstract. Moreover, probabilistic metrics (e.g., RPSS) is needed besides just simply using correlation. Given that this manuscript is not a short communication, I would encourage the authors to have a more comprehensive evaluation for SWE forecasting.*

There are many differences between the CFSv2 and GloSea5 systems that could in principle explain the higher skill of the GloSea5 system and a full answer to this question is beyond the scope of our study, but one likely reason is that the skill of the NAO/AO is higher in Glosea5 (Scaife et al 2014) than in CFSv2 (Riddle et al 2013).

We also added more details on precipitation and temperature forecasts to the abstract:

"Even though predictions for precipitation may not be significantly more skilful than for temperature, the predictive skill achieved for precipitation is retained in subsequent water balance simulations when snow water equivalent (SWE) in February is considered."

We agree that there are many more skill measures which could be addressed in our analyses. We follow your suggestion to add some more metrics. As we are using the ensemble mean instead of using individual ensemble members, we decided to use the deterministic (single value) metrics where appropriate:

- The Continuous Ranked Probability Skill Score (CRPSS) is equivalent to the Mean Absolute Error (MAE) in case of a deterministic (single value) forecast,

which is why CRPSS is used as measure representing the mean absolute error of forecasts. Here, we compute the MAE skill score (MAESS).

- The Ranked Probability Skill Score (RPSS) is equivalent to the Brier Skill Score (BSS) if a two categories forecast is considered. Thus, we will also compute the BSS values.
- We will also compute RMSE as suggested.

The revised version of the manuscript will include a table that provides these metrics. Similarly, the results of the model experiment suggested in your first comment will also analysed using these metrics. Thank you for these suggestions.

*Minor comments:*

*3. Does the AWARE water balance model distinguish the input of liquid or solid precipitation? If so, how to obtain the solid precipitation from global climate forecast model like CFSv2?*

Yes, it does. This information was still missing. We have added the method how the phase partitioning is performed in the model:

"For each grid cell the relative contributions of rainfall and snowfall are computed taking into account two threshold temperature values. If the air temperature falls below the lower threshold temperature, the monthly precipitation depth is assumed to be snowfall only. In contrast, air temperatures exceeding the upper threshold indicate rainfall only. In order to enable the occurrence of both snow and rain, a transition range between both thresholds is defined. Based on air temperature, the fraction of rain and snow is linearly interpolated between these both thresholds."

*4. What is spatial resolution for the AWARE model over the study catchment?*

[Figure]

We are sorry that is important information was also missing. Reviewer #1 also asked us to specify the spatial resolution of AWARE. It is 1000 m for the Inn headwaters.

*5. What is the definition for the benchmark Nash-Sutcliffe efficiency?*

In the revised version of the manuscript, we will provide the definition of the benchmark Nash Sutcliffe model efficiency in a new Appendix section along with the other metrics as suggested in comment #2.

*6. For the benchmark NSE during the validation period, why does it drop to 0.25? Is it because there is trend or non-stationarity in the time series?*

The benchmark Nash-Sutcliffe model efficiency is more sensitive to differences between two time series than the standard Nash-Sutcliffe model efficiency. We will explain the differences in the revised manuscript. The lower performance of the validation period might also be related to reservoir operations. We already addressed the need for a better reservoir representation in the outlook. Moreover, changes in glacier characteristics are not yet fully addressed by the water balance model. Since the calibration period was subjected to negative mass balances, positive mass balances have been observed in the 1980s. This refer to your suggestion to consider possible non-stationaries in the time series. We added the following lines to the manuscript:

Model description section:

"A possible reason for the lower $Eb$ value might be the fact that the validation period has seen an advancing of glaciers due to positive glacier mass balances. In contrast, the calibration period is characterised by a shrinkage of glaciers volumes. Both processes are not incorporated in the model so far."

Outlook:

"Moreover, a better representation of changes in glaciated area is currently being investigated through coupling AWARE with a glacier evolution model developed by Marzeion et al. (2012)."

*7. Figure 4. Besides correlation, how about the RMSE for the prediction?*

Yes, we will add RMSE as well. Please refer to comment #2.

*8. Figure 4. Is the model-simulated SWE or observed SWE used for verification? If the former, how to demonstrate the usefulness of the SWE forecasting given the limited skill in SWE simulation with AWARE (where NSE=0.25 in the validation period)?*

HISTALP-AWARE computations of SWE were used to assess the skill. Reviewer #1 also commented on model uncertainty involved in hydrological modelling. We used the reference run (HISTALP-AWARE) for verification because further uncertainties are involved in running water balance model. The revised version of the manuscript will address model uncertainty more explicitly:

"However, the comparison between HISTALP-AWARE and the CM-based seasonal forecasts highlights GCM-forecast skill and acknowledges the fact that the water balance model is never perfect since it introduces uncertainties into hydrological forecasts, too."

Please also refer to our reply to your comment #6. We added some remarks regarding possible reasons that might explain the lower model performance.

**References**

Marzeion, B., Jarosch, A. H., and Hofer, M.: Past and future sea-level change from the surface mass balance of glaciers, Cryosphere, 6, 1295–1322, doi:10.5194/tc-6-1295-2012, 2012.

Riddle, E.E., Butler, A.H., Furtado, J.C., Cohen, J.L., Kumar, A.: CFSv2 ensemble prediction of the wintertime Arctic Oscillation. Climate Dynamics 41: 1099–1116, 2013.

Scaife, A. A., Arribas, A., Blockley, E., Brookshaw, A., Clark, R. T., Dunstone, N., Eade, R., Fereday, D., Folland, C. K., Gordon, M., Hermanson, L., Knight, J. R., Lea, D. J., MacLachlan, C., Maidens, A., Martin, M., Peterson, A. K., Smith, D., Vellinga, M., Wallace, E., Waters, J., and Williams, A.: Skillful long-range prediction of European and North American winters, Geophys. Res. Lett., 41, 2514–2519, doi:10.1002/2014GL059637, 2014.

---

## Author Response (AR2)

Reply to Report #1 of Anonymous Referee #2

Dear Editor,
Dear Referee,

We would like to thank you very much for reviewing our manuscript and your constructive and critical comments. The Referee's observation regarding the re-forecasts of SWE is correct: The model experiment that includes precipitation from climatology is best in terms of RMSE even though it has higher RMSE values in case of temperature and precipitation. This might suggest that a combination of two worse forecasts, namely temperature and precipitation, results in a better SWE forecast. We are sorry that this makes the results confusing and we agree that we have to clarify our explanations in the manuscript regarding this issue.

First, we checked the computation and compared the model results with the table included in the current version in the manuscript. Since the table is in line with our results (i.e., no erroneous transfer of results to the manuscript) we also compared the results visually. In addition to the results presented in the manuscript (see, Fig. 4 in the manuscript) we prepared a similar plot (Figure 1) which includes the retrospective SWE forecasts of the three modelling experiments (full dynamical run, temperature from climatology, and precipitation from climatology).

[Figure]

Fig. 1: Retrospective SWE forecasts achieved using different modelling experiments.

Figure 1 shows the anomalies in SWE achieved in the aforementioned model experiments. These anomalies have been considered in the RMSE computations. While the time series of anomalies in Fig. 1(a) and Fig. 1(b) are similar, the anomalies in Fig. 1(c) have smaller absolute values. This observation is in line with the low standard deviation computed for the model experiment with precipitation from climatology. We already mentioned in the manuscript that the variability of SWE time series is mainly prescribed by precipitation. However, this finding alone does not explain the fact why "worse" temperature and precipitation forecasts result in better SWE forecasts. It is worth to also have a close look on the MAESS, in which errors are computed similarly but not raised to the second power as the RMSE does. In case of GloSea5 SWE forecasts MAESS is highest for the full dynamical run (0.14) even though the comparison of RMSE values would suggest that the model experiment in which precipitation is replaced has the lowest error. In turn, the lower variability combined with quadratic weighting of errors in RMSE computations could explain

that the worst combination of temperature and precipitation reforecasts are best in terms of SWE. We will discuss this effect in the manuscript:

"Surprisingly, the RMSE in terms of SWE re-forecasts is lowest in the model run in which precipitation is replaced by climatology. Since this finding is neither confirmed by comparing MAESS (which computes similar error statistics but with linear instead of quadratic weighting of errors) values nor by considering any of the other skill measures, it is likely that this effect could be explained by the low variability of SWE in this experiment combined with the quadratic weighting of errors in RMSE computations. This comparison underlines the need for different skill measures in the process of evaluating forecasts."

In the second comment in the referee's report it is argued that climate model predictions should not be worse than climatology given that downscaling and bias correction are performed properly. We agree with this statement if we consider, e.g., a re-analysis run of the climate model. However, in climate forecasting, biases might also depend on lead time. In our model experiments we consider a lead time of four months which implies additional uncertainty involved in the model runs compared to long-term climate model runs which are usually utilised in order to derive statistical measures for bias correction and downscaling purposes. Since our simple but robust approach, which is based on standardised anomalies, does not include lead time dependent adjustments, climate model-based seasonal forecasts applied in this way might be worse than climatology if we consider re-forecasts. We believe that a discussion on different downscaling methods is beyond the scope of our study. However, we consider your comment a possibility for future research. Thank you!

Some technical corrections regarding the notation of the modelling experiments were added to the manuscript.

[revised manuscript text omitted]